behaviour/cognition/psychology

decision fatigue, ego depletion, decision making, finance, risk

**Author for correspondence:**
Tobias Baer
e-mail: baertobias@web.de

At the time of acceptance, Simone Schnall was a member of the Royal Society Open Science editorial board, but had no involvement in the consideration of this manuscript.

# Quantifying the cost of decision fatigue: suboptimal risk decisions in finance

## Tobias Baer and Simone Schnall

Department of Psychology, University of Cambridge, Downing Street, Cambridge CB2 3EB, UK

TB, 0000-0003-1467-7402; SS, 0000-0002-4672-7534

Making decisions over extended periods of time is cognitively taxing and can lead to decision fatigue, which is linked to a preference for the 'default' option, namely whatever decision involves relatively little cognitive effort. Such effects have been demonstrated across a number of applied settings, including forensic and clinical contexts. Previous research, however, has not quantified the cost of such suboptimal decisions. We assessed the magnitude of the negative consequences of decision fatigue in the finance sector. Using 26 501 credit loan applications evaluated by credit officers of a major bank, we show that in this real-life financial risk-taking context credit loan approvals across the course of a day decreased during midday compared with early or later in the workday, reflecting a preference for the default option. To quantify the economic loss associated with such decision variability, we then modelled the bank's additional credit collection if all decisions had been made during early morning levels of approval. This would have resulted in $509 023 extra revenue for the bank, for one month. Thus, we provide further evidence that is consistent with a pattern of decision fatigue, and that it can have a substantial negative impact in the finance sector that warrants considerations to counteract it.

## 1. Introduction

Poor risk management in banks has caused multiple financial crises due to loan losses. Indeed, lending is at the heart of banking, with loans to small and medium enterprises (SMEs) alone amounting to over US$10 trillion [1]. For loan approvals banking personnel need to carefully weigh evidence of financial strength of the borrower against risk factors that diminish the likelihood of repayment, a taxing process that requires considerable cognitive effort. Repeatedly engaging in difficult decision making can lead to impaired or suboptimal decision making as time wears on, i.e. so-called *decision fatigue* (for a

review, see [2]). Thus, evaluating loan applications, which involves assessing the financial risks and benefits for a bank, is a cognitively demanding process for which errors can be costly to the bank.

Research on decision fatigue has demonstrated that it typically involves a tendency to revert to the 'default' option, namely whatever choice involves relatively little mental effort [3–5]. Such decision fatigue has been documented across the workday in a number of applied settings, suggesting its relevance in practical, real-life context. A seminal study showed that judges were less lenient in granting parole the more time had passed since they took a break [6]. Although methodological issues were identified for this particular study [7], similar effects were shown in other legal contexts, namely in forensic analysis [8]. In health settings decision fatigue has been documented for doctors' prescription of unneeded antibiotics [9], compliance with hand hygiene guidelines [10], and clinical decision making in surgeons [11] and in nurses [12]. Thus, considerable evidence suggests that decision fatigue can influence performance in everyday life.

It is unknown, however, to what extent default decisions typically made as a result of decision fatigue are necessarily problematic. For example, it is not clear whether rejections of parole [6] or surgeries [11] are always 'bad' decisions because there is no objective measure of accuracy that can readily be quantified. Credit approval decisions of bankers offer a unique window to examine the quality of risk decisions that may be prone to decision fatigue because there is an objective measure of decision quality: whether the loan was repaid or not. More specifically, after a bank grants a loan it is typically renewed in a yearly cycle. However, sometimes customers have difficulty making the agreed-upon monthly payment and therefore seek a revision of the contract's conditions. Such restructuring proposals pose a dilemma for the bank: on the one hand approving the request results in a loss relative to adherence to the original payment plan, but on the other hand the loss is significantly smaller than if the loan is not repaid at all. Overall banks strive to maximize the number of loans that are repaid without restructuring while minimizing the risk of loan default.[1] Risk assessment of restructuring proposals, therefore, involves binary choices of credit officers' approval (yes/no) and customers' subsequent loan repayment (yes/no), which provides an ideal context to test the quality of decisions rendered.

# 2. Material and methods

We examined 26 501 loan restructuring decisions made by a large bank's credit officers in March 2016, for which electronic time stamps indicated completed decisions.

## 2.1. Participants

Decisions made by all 30 full-time credit officers on the payroll of a bank's commercial restructuring department in the study were included in the analysis; as such, the full population of credit officers was represented. They were 11 junior analysts, seven (mid-level) analysts, eight senior analysts, three coordinators and one superintendent. The bank was unable to provide any demographic information on the credit officers other than their seniority level due to the sensitive nature of human resources data. There were a further 41 credit officers from other departments who occasionally helped out with restructuring proposals.

## 2.2. Credit decision data

The bank's decision log file from March 2016 contained 95 041 records. The majority of requests (64 950 records) had relatively straightforward risk profiles and they, therefore, were processed automatically by the system using the bank's algorithm that takes into account known risk factors. These data points were excluded from analysis. Of the remaining 30 091 requests 297 had been subsequently cancelled either by the customer or the relationship manager before a decision was made. This left 29 794 decisions processed by the credit officers. Of these, 3293 decisions (11.05%) had been made by one of 41 credit officers only helping out occasionally; these decisions were excluded because the officers probably

[1]Both a restructuring request and a failure to repay a loan are defined as a default by the central bank and therefore recorded as a breach of contract by the country's credit bureaus—this information then is shared with other banks. If a customer restructures, he or she agrees to a new repayment schedule; if this schedule is kept, banks generally maintain the relationship with the customer and after a while (typically 1–2 years) will even consider new loans to the customer. If a customer fails to repay the loan, banks will terminate all lending business with the customer and typically not consider new loans to the customer for many years.

mixed their decisions with other tasks from their regular work and therefore it was not possible to look at their performance over the course of the day. Furthermore, their lack of routine and familiarity might have introduced too much variability. Therefore, 26 501 decisions were analysed in the present study.

The decision outcome was coded as a binary variable, namely whether the restructuring proposal was approved. Rejections included outright rejections or rejections combined with a counterproposal because a distinction between the two forms of rejection was not possible due to a limitation of the bank's information technology system. Both types of rejections, however, captured the same construct, namely that the credit officer deemed the probability of the client repaying the loan to be very low.

## 2.3. Objective risk factors of loans

The bank also included specific attributes of the individual loans that it considers key determinants of risk and that it, therefore, stores in its database as part of each application's record. Loan amount, company size and a loan's delinquency status are well-established risk factors that feature in virtually every assessment of a company's credit risk (the exact procedure of which is invariably treated as highly confidential by commercial banks).

The outstanding *loan amount* ranged from $1 to over $25 million (local currency equivalent), with a median balance of $3570 (96.25% of balances exceeded $100,[2] only 1 exceeded $1 700 000). An initial analysis indicated that the approval rate was approximately linear in logarithm of amount (as opposed to nominal amount), a phenomenon routinely found with amounts as predictors of risk outcomes. Because the form of logistic regression assumes a linear relationship between the log-odds ratio of the outcome variable and the independent variables, we used the logarithm of amount for our analysis.

The *company size* was indicated by a 3-level segment indicator for small and medium company size, with large company size as the reference.

The *credit rating* is an individual bank's summary assessment of a company's credit risk and therefore a key component in calculating a bank's regulatory capital requirements, as specified by the Basel accord [13]. The rating was measured on an ordinal scale with 22 levels but as visual inspection of the data suggested a linear relationship between the rank of the rating and the approval rate, we replaced the categorical rating label with the rank, thus treating credit rating as a numerical variable.

The *collections cluster* rating is a separate metric of best practice that banks use to estimate the company's expected likelihood to pay overdue amounts [14], expressed in clusters ranging from A++ to F− (the algorithm of which is proprietary and therefore was not shared by the bank). This rating was categorical and showed no monotonous trend. The R package, which had also been used to estimate the logistic regression, was, therefore, allowed to automatically code indicator variables for each rating relative to 'A', which was chosen as the reference category by the software.

Whether or not a company had already *missed a payment* was captured through a dummy variable (1 = overdue; 0 = not overdue). The bank had hypothesized that if a loan was overdue by a larger number of days, the likelihood of approval should be higher, so we included a second variable indicating the number of days of delay. This variable was not statistically significant, however, and was removed from the final equation. One possible explanation is that this effect was already captured in a more differentiated way by the collections cluster rating.

# 3. Results

On average credit officers processed 46 restructuring proposals per day (range: 1–134, s.d. = 25.14), suggesting that a considerable number of difficult decisions had to be made relatively frequently and quickly. Only 39.95% of applications were approved; the default decision, therefore, was rejection.

## 3.1. Objective risk factors

We used a logistic regression to model the binary approval decision (yes/no) as a function of the objective loan attributes (table 1). Calculating the required sample size for logistic regression to achieve sufficient power is complex but a useful rule of thumb for the minimum sample size $N$ is

---

[2]Very small restructuring amounts can occur if a customer has two or more loans outstanding; if the larger loan requires restructuring, the smaller loan is often combined with it and thus for the smaller loan an application is also logged.

**Table 1.** Logistic regression for approval rate of restructuring proposals.

| variable | coefficient | s.e. |
| --- | --- | --- |
| logarithm of amount | −0.221*** | (0.008) |
| size | | |
| medium | −0.214** | (0.068) |
| small | −0.627*** | (0.067) |
| account not overdue | −0.731*** | (0.070) |
| credit rating | 0.034*** | (0.004) |
| collections cluster | | |
| A++ | 0.170 | (0.105) |
| A+ | 0.108 | (0.068) |
| B | −0.156** | (0.060) |
| C | −0.217*** | (0.051) |
| D | −0.323*** | (0.064) |
| E | −0.786*** | (0.053) |
| F | −1.083*** | (0.058) |
| F− | −0.858*** | (0.130) |
| decision time | | |
| 11.00–11.59 | −0.136** | (0.047) |
| 12.00–12.59 | −0.178*** | (0.051) |
| 13.00–13.59 | −0.188*** | (0.057) |
| 14.00–14.59 | −0.100* | (0.052) |
| 15.00–15.59 | 0.010 | (0.048) |
| 16.00–16.59 | −0.035 | (0.049) |
| 17.00 or later | −0.167*** | (0.048) |
| CO seniority | | |
| Level 1 (lowest) | −1.329*** | (0.094) |
| Level 2 | −1.801*** | (0.108) |
| Level 3 | −1.403*** | (0.097) |
| Level 4 | −0.357*** | (0.093) |
| officer | | |
| Number 1 | 0.256* | (0.113) |
| Number 2 | 0.186 | (0.125) |
| Number 3 | 0.015 | (0.100) |
| Number 4 | −0.133 | (0.106) |
| Number 5 | 0.453*** | (0.095) |
| Number 6 | 0.471*** | (0.103) |
| Number 7 | 0.449*** | (0.110) |
| Number 8 | 0.164* | (0.078) |
| Number 9 | 0.135* | (0.081) |
| Number 10 | 0.054 | (0.085) |
| Number 11 | 0.109 | (0.088) |
| Number 12 | −1.473*** | (0.221) |
| Number 13 | 2.265*** | (0.114) |

(*Continued.*)

| variable | coefficient | s.e. |
|---|---|---|
| Number 14 | 1.249*** | (0.105) |
| Number 15 | 0.932*** | (0.107) |
| Number 16 | 0.690*** | (0.116) |
| Number 17 | 1.458*** | (0.112) |
| Number 18 | 0.682*** | (0.115) |
| Number 19 | 0.621*** | (0.130) |
| Number 20 | 1.115*** | (0.143) |
| Number 21 | −0.581*** | (0.093) |
| Number 22 | −0.725*** | (0.124) |
| Number 23 | 1.076*** | (0.104) |
| Number 24 | 1.171*** | (0.106) |
| Number 25 | 0.983*** | (0.118) |

Significance codes: $^{***}p < 0.001$, $^{**}p < 0.01$, $^{*}p < 0.1$.

$N = 10 \times k/p$, with $k$ being the number of model parameters and $p$ being the smallest proportion of the two outcomes modelled [15]. With $N = 10 \times 51/39.95\% = 1277$, our sample was roughly 20 times larger than what would be recommended using this rule of thumb.

In credit risk management, the Gini coefficient is considered to be the best metric of a model's fit [16]; it is readily calculated by statistical packages such as R but also can be derived from the receiver operator characteristic (ROC) curve with the formula $2 \times AUC - 1$ (with AUC being the area under the curve). Our model achieved a Gini of 0.448 (corresponding to an AUC of 0.724 and therefore had a fair fit given the information available on each loan [17], comparable to the predictive power of the models this bank actually used for underwriting its loans. There were 26 451 residual degrees of freedom. Examining the model's regression coefficients, we found that the credit officers appropriately took known repayment risk factors into account: approvals were lower for larger loan amounts than smaller ones, and were also lower if the company was small, the loan was not yet overdue, the customer had a low credit rating, or the customer was assigned a lower likelihood of repayment (collections clusters A++ and A+ had the highest approval rates, clusters E, F and F− the lowest).

## 3.2. Individual baselines of specific credit officers

We also hypothesized that the credit officers' seniority level or their idiosyncratic approval propensity might affect decisions. We, therefore, introduced indicator variables for each seniority level (with the superintendent being the reference category) and additionally an indicator variable for each credit officer, with one credit officer of each seniority level randomly selected as the reference category.[3]

The four dummy variables for seniority level were all highly significant ($p < 0.001$). Overall, senior credit officers were more likely to approve a request than junior ones, thus exhibiting a lower reliance on the default option. Seventeen credit officers had a highly significant idiosyncratic effect ($p < 0.001$), eight had a weakly significant or insignificant idiosyncratic effect (four degrees of freedom were consumed by the seniority dummies, hence only 25 idiosyncratic effects remained in total).

## 3.3. Time of day for decision

To test our central hypothesis of decision fatigue we included decision time in our logistic regression that modelled binary approval decisions (yes/no) as a function of the objective loan attributes (table 1).

---

[3]We do not intend to draw any inferences about the effect of seniority on outcomes and would caution from interpreting model coefficients in such a manner, because our design implies that each 'seniority' coefficient only depicts the behaviour of one randomly chosen credit officer; we included these coefficients only for the sake of completeness.

**Table 2.** Simulated change in approval rates for an illustrative, typical application compared with approval rate before 11.00 (36.27%). The 'typical loan' was defined by the sample median for loan amount and the mode for all other attributes (due to their categorical nature); its default rate therefore is different from the average default rate across all loans.

| decision time | change in approval rate (%) |
| --- | --- |
| 11.00–11.59 | −3.08 |
| 12.00–12.59 | −4.01 |
| 13.00–13.59 | −4.22 |
| 14.00–14.59 | −2.28 |
| 15.00–15.59 | 0.24 |
| 16.00–16.59 | −0.81 |
| 17.00 or later | −3.77 |

Working times were flexible: the credit officers typically started work between 8.00 and 10.00 and finished by around 18.00, with lunch breaks typically taken somewhere between 13.00 and 15.00. The electronic system only logged the time when a decision was registered rather than whether a credit officer engaged specifically in credit assessment or other clerical tasks, or when they took a break. We defined seven time indicators for discrete time periods, over the course of which we expected decision fatigue to occur, with six dummies indicating each respective working hour between 11.00 and 17.00 and one dummy indicating a decision time at 17.00 or later. The dummies hence compared approval propensities to decisions taken before 11.00.

We tested whether the default decision of rejection would be more likely by midday (i.e. when the credit officers reached their lunch break), and lower after they probably had taken such a break. Indeed, relative to early in the day, approval rates were significantly lower for each time slot between 11.00 and 14.00. Coefficients were −0.136 at 11.00–11.59 ($p = 0.003$), −0.178 at 12.00–12.59 ($p < 0.001$) and −0.188 at 13.00–13.59 ($p < 0.001$), with no significant differences between those individual time slots. There furthermore was a return of approval rates later in the workday, with coefficients not significantly different from 0 between 15.00 and 16.59 (absolute coefficient values less than one standard error). After 17.00 we observed another significant drop in approval rates, as indicated by a coefficient of −0.167 ($p < 0.001$). Thus, consistent with the notion of decision fatigue occurring after multiple working hours [18], it was observed in the late morning and late afternoon hours.

The logistic regression ($\chi^2 = 4088.4$, $p < 0.001$) also allowed us to simulate the change of approval probability for a particular application depending on the time of day (table 2). We modelled how many additional restructuring requests in our sample would have been approved had they been assessed at the time of peak performance (i.e. before 11.00). The simulation suggests that on the margin, 569 additional loans would have been approved, thus increasing the overall approval rate from 39.95% to 42.09%.

## 3.4. Cost of decision fatigue

Our second research question concerned the economic cost of decision fatigue to the bank. Because we also had data for loan repayments in the subsequent 12 months, we were able to link up each credit decision with an objective outcome measure, namely whether customers subsequently repaid any due portions of the loan. For approved restructuring requests the repayment rate was 52.62%, whereas for rejected it was only 38.71%. As a next step we explored whether the default decision of rejecting a request was associated with a disadvantageous financial outcome for the bank.

Out of the 569 marginal restructuring requests that we had identified above, 40.42% had been repaid even though the restructuring request had been rejected. Critically, we can then ask the key question regarding decision quality, namely how many more loans would have been repaid had the restructuring request been approved. We built a logistic regression to predict the probability of repayment as a function of each loan's objective attributes plus a binary variable indicating restructuring approval (yes/no). This model (table 3, $\chi^2 = 11\,237$, $p < 0.001$) showed that whether the restructuring proposal had been approved was a highly significant factor influencing repayment of the loan ($p < 0.001$), along with the objective risk attributes loan amount, company size, delinquency status, credit rating and collections cluster.

**Table 3.** Logistic regression for successful repayment of overdue amount.

| variable | coefficient | s.e. |
|---|---|---|
| restructuring approved | 0.392*** | (0.033) |
| logarithm of amount | 0.185*** | (0.009) |
| size | | |
| medium | 0.427*** | (0.062) |
| small | 0.239*** | (0.057) |
| account not overdue | 0.443*** | (0.071) |
| credit rating | 0.120*** | (0.004) |
| collections cluster | | |
| A++ | 0.252* | (0.121) |
| A+ | 0.152* | (0.077) |
| B | −0.066 | (0.067) |
| C | −0.554*** | (0.056) |
| D | −1.622*** | (0.067) |
| E | −3.296*** | (0.065) |
| F | −3.962*** | (0.080) |
| F− | −3.983*** | (0.205) |

Significance codes: ***$p < 0.001$,** $p < 0.01$, *$p < 0.1$.

Our simulation of the number of additional approvals follows the typical approach of how banks apply decision scorecards. Our logistic regression estimates the probability of approval of each application in the sample; an application is assumed to be approved if the probability of approval reaches a critical value (cut-off score). In the sample 39.95% of applications were approved; we, therefore, set the cut-off score at the 60.05th percentile (=100–39.95%) of the score distribution, which was 0.0827. In other words, 39.95% of observations in the sample had a score greater than 0.0827, and we, therefore, assume that any application with a logistic score greater than 0.0827 is approved. If we evaluate each application as if it had been decided before 11.00, the number of applications above the critical value rises from 10 586 to 11 604, an increase of 1018 applications. Of these, however, 449 applications already had been approved—this is because the logistic score is not a perfected classifier and therefore any logistic regression will misclassify a portion of the observations; specifically, some accounts that actually had been approved were misclassified by being assigned a score below the critical value. Taking these 449 already approved applications into account, we can conservatively assume that an incremental 569 applications would have been approved (figure 1).

To further corroborate this estimate, we compared this precise, bottom-up estimate with an alternative, top-down estimate. Table 2 had shown the change in approval rate for a *typical* application during different periods of the day. Considering the number of applications decided in each period and applying this difference in approval rate, we would estimate an incremental 522 approvals assuming that all applications have the 'typical' profile, as can be seen in table 4.

In order to estimate the incremental number of loans that would have been repaid if all decisions had been taken before 11.00, we twice scored each of the 569 applications identified in the logistic regression estimating the probability of successful repayment of the overdue amount (table 3)—first setting approval to 'yes', and then to 'no'. We thus assessed the increase in the probability of repayment for each of the 569 loans—e.g. if the probability of repayment rose from 10% to 15%, we registered an increase of 5% points. By adding up the differences in these loan-level probabilities, we determined the total number of loans that would have been repaid—e.g. for 100 loans with an increase in the probability of repayment of 5% points each, we would estimate $100 \times 0.05 = 5$ loans additionally repaid (i.e. 5%). For the 569 loans, this calculation added up to 32 loans. We then estimated the incrementally repaid amount by multiplying each loan's increased repayment probability with the loan amount, adding up the incremental expectation in recovery for each of the 569 loans. This yielded an amount of $509 023.

**Figure 1.** Simulated approval rates for an illustrative, typical application. The 'typical loan' was defined by the sample median for loan amount and the mode for all other attributes (due to their categorical nature); its default rate therefore is different from the average default rate across all loans.

**Table 4.** Simulated change in the number of approvals by changing decision time interval to a period with less decision fatigue.

| decision time | cases handled | change in approval rate (%) | incremental approvals in absence of fatigue |
|---|---|---|---|
| 11.00–11.59 | 3629 | −3.08 | 112 |
| 12.00–12.59 | 2771 | −4.01 | 111 |
| 13.00–13.59 | 2064 | −4.22 | 87 |
| 14.00–14.59 | 2654 | −2.28 | 60 |
| 15.00–15.59 | 3231 | −0.24 | — |
| 16.00–16.59 | 3030 | −0.81 | 25 |
| 17.00 or later | 3367 | −3.77 | 127 |

## 3.5. Potential confounds

The study of Israeli judges' parole decisions by Danziger *et al.* [6] has been challenged by the suggestion that judges might have ordered the cases according to difficulty [19]. In addition, a simulation of these data demonstrated that if parole approvals took longer than rejections, a statistical artefact may inflate the effect ascribed to decision fatigue [7]. In the following, we demonstrate that neither concern is applicable to our study.

### 3.5.1. Case ordering

The bank indicated that the sequence in which decisions are made is driven by an electronic workflow system and hence cannot be influenced by credit officers. There are multiple work queues that are processed in sequential order—small/medium/large cases are randomly assigned among all credit officers with junior/medium/senior seniority. Excess volume (load balances) is picked up by other credit officers in sequential order; cases referred to more senior credit officers due to complexity are also actioned sequentially. Therefore, all work queues amount to a random assignment. In addition, we tested whether the time of the day had a significant impact on the loan repayment rate. An effect of decision time would suggest that there are systematic differences in the risk profile of the applications depending on the time of day. We, therefore, added the indicator variables for different periods of the day to the logistic regression estimating the probability of successful repayment. As shown in table 5, the time indicator variables were not significant, supporting the hypothesis that no systematic ordering of applications had taken place. In fact, the only time period with a weakly significant effect (15.00–15.59, coefficient −0.126, $p = 0.024$) was a lunch hour during which we did not find any evidence of decision fatigue. The data, therefore, offer little support for the hypothesis that the cases decided during midday may be different in some way from normal cases.

**Table 5.** Logistic regression for successful repayment of overdue amount, controlling for time of decision.

| variable | coefficient | s.e. |
| --- | --- | --- |
| decision | | |
| 11.00–11.59 | 0.062 | (0.054) |
| 12.00–12.59 | 0.018 | (0.059) |
| 13.00–13.59 | 0.027 | (0.065) |
| 14.00–14.59 | −0.098 | (0.060) |
| 15.00–15.59 | −0.126* | (0.056) |
| 16.00–16.59 | 0.047 | (0.057) |
| 17.00 or later | 0.026 | (0.056) |
| restructuring approved | 0.394*** | (0.033) |
| logarithm of amount | 0.185*** | (0.009) |
| segment | | |
| medium | 0.428*** | (0.062) |
| small | 0.242*** | (0.057) |
| account not overdue | 0.449*** | (0.071) |
| credit rating | 0.120*** | (0.004) |
| collections cluster | | |
| A++ | 0.255* | (0.121) |
| A+ | 0.154* | (0.077) |
| B | −0.065 | (0.067) |
| C | −0.556*** | (0.056) |
| D | −1.626*** | (0.067) |
| E | −3.301*** | (0.065) |
| F | −3.965*** | (0.080) |
| F− | −3.987*** | (0.205) |

Significance codes: $^{***}p < 0.001$, $^{**}p < 0.01$, $^{*}p < 0.1$.

### 3.5.2. Time spent per case

To address the concern that time spent per decision might have differed for approvals and rejections [7], we measured the time passed since the previous decision made by the same credit officer and compared the measurements for approved and rejected decisions. There was indeed a significant difference, two-sided $t$-test (conservatively assuming unequal variance): $t = 7.126$, $p < 0.001$ (approvals: mean = 12.9 min, s.d. = 19.1, minimum 0, maximum 316 min; rejections: mean = 11.2 min, s.d. = 17.1, minimum 0, maximum 359 min). While both the absolute time saving (rejections were 1.7 min faster) and the relative time saving (13%) were less than what Glöckner [7] had observed in the parole decision sample (2.2 min and 29% time saving, respectively), this significant difference still raises the question whether credit officers systematically might have postponed approvals until after lunch, thus creating an artificial effect inflating the reject rate among decisions taken before lunch. We investigated this hypothesis by testing the significance of the length of time spent on each case as a predictor of approval alongside the time of the decision. As shown in table 6, however, the time spent on each case (case length) was *not* significant (coefficient 0.001, $p$-value 0.145), whereas the decrease in approval rates observed from 11 o'clock onwards continued to hold (coefficients of −0.119, −0.165, −0.178 and −0.088, with $p$-values of 0.013, 0.002, 0.002 and 0.097, respectively, and for the hourly intervals commencing at 11, 12, 13 and 14 o'clock, respectively).

## 4. Discussion

Results from the current investigation show that in a real-life context, namely in the finance sector, extended periods of cognitive exertion are consistent with a pattern of decision fatigue: while early in

**Table 6.** Logistic regression for approval rate of restructuring proposals as per table 1 but with case length as additional variable.

| variable | coefficient | s.e. |
|---|---|---|
| case length | 0.001 | (0.001) |
| decision time | | |
| 11.00–11.59 | −0.119* | (0.048) |
| 12.00–12.59 | −0.165** | (0.052) |
| 13.00–13.59 | −0.178** | (0.057) |
| 14.00–14.59 | −0.088* | (0.053) |
| 15.00–15.59 | 0.025 | (0.049) |
| 16.00–16.59 | −0.020 | (0.050) |
| 17.00 or later | −0.158** | (0.049) |

Significance codes: ***$p < 0.001$, **$p < 0.01$, *$p < 0.1$.

the morning approval rates for credit risk applications were high, they declined toward the middle of the workday, then rose again later in the afternoon. More specifically, during times of suboptimal performance credit officers were more likely to revert to the default decision, namely to reject the loan application. This work, therefore, adds to the growing evidence that in ecologically valid and practically significant situations decision quality can suffer [6,8–12].

Going above and beyond this earlier work, however, the current study quantified the extent to which decision quality was inferior. Indeed, in some contexts making default decisions may not always be disadvantageous. For example, in the case of parole decisions [6], antibiotic prescriptions [9] and clinical decision making in healthcare professionals [11,12], being cautious might be a benefit. Similarly, given the high financial stakes, for credit loan approvals 'being on the safe side' and therefore rejecting a request might be the preferred course of action.

Because we had information about subsequent repayment of loans we were able to probe this question. The results showed that rejection of restructuring requests (i.e. not changing a customer's contract to facilitate repayment) made it less likely that a loan was subsequently repaid, therefore, clearly indicating that decision fatigue was associated with negative outcomes. Indeed, the financial loss of decision fatigue was more than 9 times the average monthly salary of individual credit officers,[4] suggesting that in the banking sector—and possibly in other work contexts—the economic costs of cognitive depletion can be substantial. Moreover, we only examined one month of decisions, so the financial loss may be greatly amplified over the long run.

This research also speaks to the contentious research area of ego depletion[5] [22], for which questions have been raised regarding the extent to which such effects are consistent and reliable [23–25]. Our findings support the idea that depletion of self-regulatory processes may occur only over extended periods of time. Indeed, while brief laboratory manipulations have sometimes failed to show depletion effects, they have been observed when participants had to engage in a cognitively taxing task over extended periods of time [18]. Our work, therefore, fits with calls to explore a range of methods to test the implications of reduced self-control [20].

A limitation of the current work is that although we identified patterns indicative of decision fatigue across the day, we were unable to pin down the precise timing of breaks, therefore, making it difficult to

[4]The benchmark salary was conservatively derived as the maximum senior credit officer compensation indicated by the first three salary benchmark websites in the banking sector of the bank's country obtained by a Google search on 19 May 2017, which we deemed to be a fair reflection of the salary levels in force in March 2016 (i.e., during the time of data collection). The second-highest source was about 10% lower than the source with the highest benchmark; the third source did not distinguish between seniority levels and had a surprisingly low benchmark for credit officers in general that was only half of the highest benchmark of junior credit officers and less than a third of the senior credit officer benchmark we studied. We refrain from naming the actual websites in order to protect the identity of the bank but want to indicate with this elaboration that our statement that the financial loss from decision fatigue is more than 9 times the average salary is very conservative and likely to understate the true relative scale of the damage.

[5]Definitions of the concepts of ego depletion, self-regulation and cognitive control have received extensive attention (e.g. [20,21]). These theoretical discussions are outside the scope of the current investigation, for which we do not take a strong theoretical stance.

demonstrate a direct link between breaks and the increase in approval rates in the afternoon, an effect we hypothesize to reflect the recovery from ego depletion. Regardless of the precise mechanism, however, given the high stakes involved in credit approvals across the worldwide banking sector, our findings have important practical implications.

Our research provides a strong impetus for banks to take measures to reduce decision fatigue. Indeed, established methods involve counteracting decision fatigue by making effortful activities less depleting (e.g. by brief rest periods), an idea that is consistent with suggestions that mental fatigue and physical fatigue share a common physiological substrate [26]. Another possibility would be to offer small incentives for sustained self-control specifically during periods of fatigue [27]. Furthermore, a recent meta-analysis showed that self-control is a skill that can be improved by training [28], and this might have relevance to the work place. Thus, although modern work patterns have been characterized by extended hours and higher work volume, cutting down on prolonged periods of intensive cognitive exertion may be more productive for banks and borrowers alike.

Ethics. The authors confirm that ethical guidelines for archival data analysis, as outlined by the University of Cambridge were followed.

Data accessibility. This study includes data received from a commercial bank. These data involve highly sensitive personal information about customers (e.g. credit history; prior repayment history of loans). They are furthermore proprietary and would provide a substantial competitive advantage to other entities if made publicly accessible, and if the name of the bank was revealed. T.B. was asked to sign a non-disclosure agreement before the data were communicated to him, and this unfortunately prevents the authors from making the dataset publicly available. This has been agreed with the journal's editors. However, we provide an illustrative dataset to show the structure of the dataset, its data elements, and the nature of the values in individual fields, so readers can evaluate the analytic strategy behind the findings. This illustrative dataset is available as electronic supplementary material.

Authors' contributions. T.B. and S.S. conceived the research question and designed the study. T.B. obtained and analysed the data. T.B. and S.S. wrote the manuscript. Both authors approved the final version of the manuscript.

Competing interests. At the time of writing, Dr Simone Schnall was a Board Member of Royal Society Open Science, but had no involvement in the review or assessment of the paper.

Funding. The authors declared no funding.

Acknowledgements. T.B. thanks Andreas Richter for advice during the data analysis and interpretation. The authors also thank the bank that provided the data for this research.

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
