## [Peer Review File · Royal Society Open Science]

Review History

RSOS-201059.R0 (Original submission)

Review form: Reviewer 1

Is the manuscript scientifically sound in its present form?

Yes

Are the interpretations and conclusions justified by the results?

Yes

Is the language acceptable?

Yes

Do you have any ethical concerns with this paper?

No

Have you any concerns about statistical analyses in this paper?

No

Recommendation?

Major revision is needed (please make suggestions in comments)

Comments to the Author(s)

This paper studies the decisions on credit restructuring applications in different time frames in a commercial bank. Authors use it to test the cost of decision fatigue, with the hypothesis that when decisions are taken along the day, decision fatigue drives to choose the default option. Moreover, the authors quantify the costs that decision fatigue has for the bank.

The research questions are important and well defined according to the literature. The statistical analysis, in general, is adequate and allows to answer the questions. I consider that the paper is very interesting, well written and makes a very nice contribution to the literature. I also consider that there are some points that need to be clarified.

Major points:

- Authors propose that they are the first ones that show that decision fatigue produces bad outcomes. I think that it is not exactly so. It is implicit in the previous studies that choosing more often the "default option" is a bad outcome. Authors argue that it is not the case that the "default option" is a "bad option". Although it is true, it is also true that choosing too often the default option is a bad option. For instance, Persson et al (2019) finish their paper arguing that "From a societal point of view, this is an inefficient and arguably unfair use of medical resources" in their analysis of decision fatigue in the context of surgeons' clinical decision making. Inefficient means here that it is "bad". On the other hand, in this paper the author are the first who quantify the cost, and I think that this certainly is important and deserves credit. I consider that authors should state better that they are the firsts who quantify the cost of decision fatigue rather than the firsts who show that decision fatigue IS costly.

- My second major concern is with respect to the Analysis in Table 6 regarding Time spent per case. As far as I understand the analysis, the authors argue that they extend the logistic regression 1 adding case length as an additional explanatory variable (they described it in that way in the name of the table). If it is the case, the authors should include the coefficients of all the independent variables as in Table 1. Moreover, and it seems strange, they merge in the set of explanatory variables in Table 6 the dummy variable 16:00-17:00 and 17:00 or later (Table 1) in the variable 16:00 or later (Table 6). I don't see the reason, I suppose it is only a mistake.

Minor points:

- In Materials and Methods, I would prefer if the country of origin of the bank is included. Maybe it cannot be said because of privacy, if it is the case it should be disclaimed. In Footnote 2 authors describe wages in the sector in the country, I suppose that from those data country can be inferred, so it would be better to show it clearly (or to delete these wages).

- The Results section would be enriched with some descriptive statistics. In particular, given the objectives of the paper and the analysis, approval rates by time framing seem a natural description of the data.

- In Results -> Individual baselines of specific credit officers. I am not sure on what are doing the authors here. As I understand, they are adding control variables to regression 1. However, if those control variables are already included in regression 1, it should be explicitly explain in Table 1. If Table 1 is not controlling by those factors, then the new regression should be included somewhere, in order to check if coefficients in Regression 1 are robust or not. If that regression

has not been included, given that its importance is only marginal, I suggest to include it as an Appendix.

- Results -> Time of Day for Decision. Since this paragraph is included after the previous one and it refers to Table 1, I cannot know if "Seniority" is controlled for these results or not. Please clarify.

- When the authors compute the repaid amounts, they assume that this is a lower bound estimation of the cost that fatigue has for the bank. However, typically, when the bank accept a restructuration, te bank assumes some costs (for instance, accepting an extension of payments or similar). Since the restructuration costs are not taken into account in the calculation, the estimated costs are not really a lower bound. The authors should discuss the magnitude of these expenditures for the bank. Is there any estimation of how costly is for the bank to accept a restructuration?

Review form: Reviewer 2

Is the manuscript scientifically sound in its present form?

Yes

Are the interpretations and conclusions justified by the results?

Yes

Is the language acceptable?

Yes

Do you have any ethical concerns with this paper?

No

Have you any concerns about statistical analyses in this paper?

No

Recommendation?

Accept with minor revision (please list in comments)

Comments to the Author(s)

See attached file (Appendix A).

Review form: Reviewer 3

Is the manuscript scientifically sound in its present form?

Yes

Are the interpretations and conclusions justified by the results?

Yes

Is the language acceptable?

Yes

Do you have any ethical concerns with this paper?

No

Have you any concerns about statistical analyses in this paper?

Yes

Recommendation?

Major revision is needed (please make suggestions in comments)

Comments to the Author(s)

The article "Quantifying the Cost of Decision Fatigue: Suboptimal Risk Decisions in Finance" investigates if loan officers' decisions to approve loan restructuring are affected by decision fatigue. In addition, the study analyses potential consequences of the decision fatigue. The study uses a dataset from one of the major banks including 26 501 decisions of 50 loan officers made throughout one month. The results are in line with decision fatigue and, furthermore, decision fatigue has negative effect on the bank's revenue. I think that the paper is well-written, the research methods are appropriate and the literature review complete. This review details some concerns I have about the study that I suggest being addressed before a decision about publication can be made.

Major comments:

- Authors define "default option" as the option that requires the least cognitive effort. However, this is not a precise definition of a default option. I do agree with the authors that not approving the application is a reasonable default, however, I would like to see more arguments speaking for it. After all, approving the application may be seen as "safer", i.e., the restructured loans are more likely to be repaid.
- In the "sample size considerations" authors mention that they have sufficiently big sample (26 501 observations) to detect small-size effects. However, it is not clear whether they took into account the correlation between observations of the same decision-maker. In other words, in simple power calculations it is assumed that the observations are independent of each other, however, given the nature of the dataset it is safe to assume that decisions of the same loan officer will be correlated. If these correlations are relatively big, bigger samples are needed to detect the same effects. Having said that, I do agree that the sample is big enough and the power of conducted tests should not be a problem. While it is tricky to determine sample size for longitudinal data (if the clusters are of roughly equal size the formula for design effect can be used), I would suggest authors to at least specify whether they accounted for the correlation between observations and point out how this can affect the sample size calculations.
- The manuscript would benefit from better description of the dataset and the handling process in the Materials and Methods section. Information that would be of interest is: how are the cases assigned to different employees, is the order of cases random, etc. Authors describe very briefly in the Results section under "Case ordering" that cases are not ordered by loan officers, but I think it would be beneficial to describe it early on in the paper for the readers to understand the results and study design.
- The authors try to tackle the problem of potential confounds. I find the analysis conducted to test whether time spent on the case mattered for the approval decision a valuable robustness check. In addition, authors also try to test if case ordering mattered for the repayment success. If there are specific patterns in the order of considered proposals, they can be the reason for the found time-of-the-day effect, not the decision fatigue. However, in Table 5 the explanatory variable "restructuring approved" may already pick up the variance related to time of the day, as these approvals are time-dependent. Another way to show that the ordering of cases was time-independent would be to test (e.g., in a regression setting) if the characteristics of the applications (i.e., loan amount, company size, credit rating, collections cluster and missed payment) change over time. These results could be kept in an electronic supplementary material.

- Why do the time-variables start from 11am rather than from 09am/10am? While I understand the approach, it would be interesting to see the explanation from the authors. Maybe to make it clearer it would be useful to add a graph that shows the fraction of approved cases throughout the day (starting from 08am)?
- "We also hypothesized that the credit officers' seniority level or their idiosyncratic approval propensity might affect decisions." I would suggest to move the hypothesis to methods section, where authors describe the remaining predictions. In addition, it is not entirely clear to me whether the indicator variables were introduced in the logistic regression (it is not shown in Table 1) or were tested separately. Furthermore, given that there were 50 decision-makers and 5 levels of seniority, the conclusions regarding seniority should be made with caution
- I would like to see a discussion on the caveats related to the method used to estimate the impact of decision fatigue on bank's revenue. For example, authors use variable "restructuring approved" (Table 3) as exogenous variable in the regression estimating the success of the loan repayment. However, the approval depends on the characteristic of the case. In other words, it may be that loan officers indeed approved the applications that wouldn't have been successful otherwise and rejected the applications that would have been unsuccessful anyway. In that way the approval would increase the likelihood of repayment, but does not necessarily mean that that other applications that were rejected would have been successful if they were approved.

Minor comments:

- a typo in the title: "suboptimal" instead of "supoptimal"
- authors use "credit loan approval" interchangeably with "loan restructuring approval". In my opinion these two terms describe different things and may be misleading for some readers. Thus, I would suggest being clearer/more explicit in the use of different terms and, for instance, stick with the term "loan restructuring approvals"
- the first paragraph in the section "Time of Day for Decision" would fit better in the Materials and Methods section. It makes it clearer how the work of credit officers looks like, when they take breaks etc. which is of interest in the description of methods to better understand the design of the study.
- In the logistic regression in Table 3 authors model success which is defined as "repayment of overdue amount". However, some of the companies did not have any amount overdue and authors even use "account not overdue" as one of the explanatory variables. I assume that this is just a matter of description, but it would be good if authors better describe what they mean by "success" in Table 3
- The description of the analysis of the consequences of decision fatigue on bank's revenue is difficult to follow due to its organization. For instance, authors mention on page 4 that 569 additional loan restructuring applications would be approved but they describe the method used to find that value on page 5 in rows 21-32. This description should be moved to the last paragraph on page 4.
- On page 5/13 in rows 33-37 authors present an alternative calculation for the number of approved applications. However, they do not comment on it or use it in the future calculations. It is not clear what is the purpose of these additional calculations.
- missing reference for Vohs et al. 2008
- references are not in alphabetic order

Review form: Reviewer 4

Is the manuscript scientifically sound in its present form?

Yes

Are the interpretations and conclusions justified by the results?

No

Is the language acceptable?

Yes

Do you have any ethical concerns with this paper?

No

Have you any concerns about statistical analyses in this paper?

Yes

Recommendation?

Major revision is needed (please make suggestions in comments)

Comments to the Author(s)

In this article, the authors analysed a data set of credit officers working for a bank and deciding to approve or not loan restructuring requests. This decision involves a trade-off because approving a loan comes with financial loss but increases the chance of avoiding bigger financial loss if the company does not repay, but not approving decreases the likelihood for the company to repay. Using a very rich data set (30 credit officers working for one month, resulting in 26,501 decisions), the authors show that objective factors (i.e., objective risk attributes) as well as the time of the day (presumably decision fatigue) are influencing credit officers' decisions. They also compute the financial loss resulting from the time of the day influence on officers' decisions. These are important findings, potentially showing the real-life consequences of decision fatigue.

I however have some concerns that should be addressed as well as suggestions for improvement.

Major

Regarding data availability:

1) I am uncomfortable reviewing an article using a data set that is not available in any way. There is no need to specify the name of the commercial bank, and no need to indicate the name of customers. The problem with data that cannot be shared are manifold, one of them has been illustrated during the covid-19 crisis, with a paper based on a confidential data set regarding the use of hydroxychloroquine to treat patients. Alternatively, perhaps the name of the bank could be mentioned, opening the possibility for someone aiming to replicate the analysis to do so?

Regarding the method:

2) Methods are too concise, and the analyses and simulations are too vague. The analyses that have been used are described in the result section but should also be described more thoroughly, in the methods section, indicating clearly the parameters which were used and spelling out the equations.

3) The effect of seniority and the effect of time of day corrected for objective risk attributes are estimated in different regressions. A linear mixed model can be used to assess all the effects together as well as the potential interactions (e.g., junior officers would be more prone to decision fatigue).

Regarding the results:

4) The main effect (time of day on decision) is supported by a unique analysis. I suggest the author do use different strategies to further support the results. For example, I suggest running 30 logistic regressions (one per credit officer) with the dummy variables for each time of day and the regressors describing the objective risk attributes to then test the parameters at the group level against the null distribution. This would be accounting for individual difference between officers. This could also be done for each day separately if for example the day of the week affects decisions.

- 5) The data should be shown graphically. More specifically, the requests acceptance rate should be plotted by time slot, with standard error as error bars. Then, the model prediction should be overlaid so that the goodness-of-fit can be evaluated. I also suggest plotting the averaged residuals by time of day of 30 regressions with the objective risk attributes as regressors. This would correct the time series for the objective risk – which is a confound here.
- 6) Perhaps the time of day could be fitted to the decisions as a parametric regressors (e.g., encoded as 1, 2, 3, etc.) instead of using many dummy regressors? The latter is agnostic regarding the shape of time of day effect on decisions, while the former assume a linear effect.
- 7) More details about the statistics should be reported (e.g., average of the estimate, uncertainty about the estimates, t-value, degree of freedom) besides the p values. The authors should also report goodness of fit metrics, like the R2 or pseudo-R2 as well as the balanced accuracy of the model.
- 8) Many tests are used for each dummy variable. Applying a correction for multiple comparisons should be considered.
- 9) Cognitive fatigue has two impacts on the financial loss, to my understanding:
- a) A “negative cost”, i.e. a benefit, corresponding to the difference between restructuring and not restructuring the 40.42% of companies that repaid. If these restructuring requests would have been accepted, this would have been detrimental as they would have paid less than what they actually paid.
 - b) A cost that corresponds to the product between the increase in the likelihood of repaying given the restructuring approval and the amount that would be repaid. The increase in the likelihood of repaying given the restructuring approval should be corrected for objective risks attributes, which are different between the approved and disapproved requests (because it influences the decisions as reported by the authors). In fact, the improvement is not simply the difference in repayment between the approved and rejected requests as it could be interpreted from lines 6-8 page 5 (“For approved restructuring requests the repayment rate was 52.62%, whereas for rejected it was only 38.71%. Thus, the default decision of rejecting a request was associated with a disadvantageous financial outcome for the bank.”). One could assume that the objective risk difference between the approved and rejected requests may fully explain the difference in the repayment proportion.
- The logic behind the analyses page 5 is hard to follow, because it is not explicitly described. Also, to my understanding, not all the information that is presented is necessary to compute the financial cost of decision fatigue in that context and all the necessary information is not presented. I therefore suggest using analyses based on the aforementioned cost definition.
- 10) The lack of control group and the lack of information regarding the time and the duration of breaks are limitations that the authors mentioned. They may discuss alternative hypothesis, for example the effect of circadian rhythms, or perhaps a prior estimate of the number of acceptance rate per day. In fact, the data cannot support the idea of decision fatigue. I suggest using “time of day” as much as possible and to offer decision fatigue as a main interpretation.

I also have minor suggestions:

- 11) I suggest using cognitive fatigue instead of ego-depletion as the latter has been deeply criticised if not invalidated (as mentioned in the discussion), while the former has been used for about a century and less criticised.

- 12) The main questions, the main metrics that will be used , as well as the predictions are not explicitly described in the introduction.
- 13) Does the day of the week influence the decisions?

Review form: Reviewer 5

Is the manuscript scientifically sound in its present form?

Yes

Are the interpretations and conclusions justified by the results?

Yes

Is the language acceptable?

Yes

Do you have any ethical concerns with this paper?

No

Have you any concerns about statistical analyses in this paper?

No

Recommendation?

Major revision is needed (please make suggestions in comments)

Comments to the Author(s)

The present manuscript describes an interesting study that examines the consequences of decision fatigue on financial decision making, specifically credit loan applications. The authors hypothesize that credit officers will be more likely to approve applications when they are less fatigued (in the morning), but will default to reject more applications when they experience greater fatigue. The most intriguing aspect of this research is the data set that the authors were able to access to conduct their work. Namely, they were given access to a large bank's data for the month of March 2016 in which over 26,000 relevant credit loan applications were processed. Examination of these data revealed a pattern consistent with previous work on decision fatigue, illustrating the predicted trend in greater tendency to approval applications in the morning, and a greater rejection rate in the midday and late afternoon. Perhaps the most critical aspect of the present work is that the authors were able to explore whether these decision fatigue effects actually had deleterious consequences for the bank and its decision makers. By extrapolating from the repayment rate, the authors were able to show that the tendency to reject applications due to decision fatigue cost the bank a considerable amount of money. In this way, the authors tout this study as the first to illustrate the negative downstream consequences for decision quality resulting from fatigue/depletion.

As a researcher actively engaged in work relevant to depletion effects, I read this manuscript with great interest. Certainly, this paper builds nicely on recent efforts that show depletion effects in real-world decision making contexts, such as judges' parole decisions (Danziger et al., 2011) or doctors' prescriptions of antibiotics (Linder et al., 2014). The authors examine the same hypothesis in a new setting, focusing here on the financial decision making of bank employees processing credit loan applications. Consistent with the past work, the authors expect that credit officers will be more likely to take the risk of approving credit loan applications when they are

less fatigued, but will follow the default of being conservative and reject credit loan applications more when they are fatigued. The results seem to largely support this hypothesis, though the data leave something to be desired. That is, given the nature of data which the authors have to work with, they were unable to track the fatigue levels of the credit officers as neatly as some past studies have. For instance, in the Danziger et al. work, they were able to track fatigue levels throughout the day by noting when the decision makers (judges) took their breaks during the day. Specifically, they noted that after breaks (such as lunch), there was a temporary restoration of behavior corresponding presumably to reduced levels of fatigue, resulting in a scalloped shape function throughout the day. In the present research, without access to the specific times at which credit officers took their lunch (or other) breaks during the day, the authors had to simply extrapolate when most people took their lunch breaks to infer the presumed fatigue functions for these decision makers. These limitations are simply unavoidable, given the limitations imposed on those granting them access to the present data set, but they do limit the ability to draw clear conclusions about the role that fatigue presumably plays in the observed function of credit approvals. We can however safely assume that people are at their “best” (or at least are less fatigued) at the beginning of their work day and fatigue tends to increase as the day wears on.

Aside from these limitations in terms of the presumed fatigue levels of the decision makers, the ability to draw conclusions about the financial consequences of these decisions relies heavily on projections based on the banks’ historic data of repayment rate of loans. This seems like a fair way to project the potential financial costs of decision fatigue here, but I will admit that I would have loved to have seen a more nuanced analysis of the applications that did or did not get approved as a function of time of day. I know the authors were not able to parse the data in this way, but it would be very informative to see what parameters of the loan applications get prioritized or ignored when decision makers are or are not fatigued. The authors have access to credit rating, loan amount, seniority level of the credit officer, but from a pragmatic standpoint, I was curious as to the potential for other factors (like SES or race) to infiltrate decisions at this point. That is, the authors conjecture that decision makers opt to follow their defaults more when fatigued, and that the default in this case is to be conservative and reject the application. But in other decision making contexts, we have seen that many other defaults can be operative when people are fatigued or stressed or distracted. People tend to rely on cultural stereotypes more, and thus may reject applications from minority group members (or companies or organizations) when fatigued than when at their full capacity. I know the authors are not able to explore these possibilities, but it is the case that there are other defaults that may be operative, if and when decision makers are feeling fatigued, that may govern their judgments and decisions under these conditions. It may not be that there is a drop in approval rate of any or all applications, but only of certain applications. And this point becomes particularly salient when the authors note that they could not control whether the credit officers look at these applications in a random order or organize the applications in some way (such as doing the easier ones first and saving the more difficult ones for later, or vice versa).

All this said, I think the present work is competently done and provides an intriguing new context in which to study the fundamental question of the (negative) consequences of decision fatigue. What seems to be an emerging pattern in studies like this is that decision fatigue (or depletion) effects appear to be more robust in real world settings, where you have real decision makers doing activities over time that cause fatigue. Thus, these studies have been helpful in reinforcing faith that depletion effects exist in the real world, despite the challenges (and failures to replicate) that researchers have observed in lab studies of this phenomenon. I really like the exploration of these questions in this particular financial decision making context, and sincerely appreciate the value of the present data set (with all its richness and external validity, despite the obvious and unavoidable limitations) to serve as the basis for this investigation. So from that perspective, I am very favorable disposed toward the publication of the present work. Still, though, the question to me is the level of contribution of the present work, over and above past

real world demonstrations. The authors tout that this investigation affords the possibility of assessing the actual financial consequences of decision fatigue for the bank (in terms of profits or income). I appreciated their efforts in this regard, but I found that argument a bit hollow. The earlier studies which examined health care professionals' compliance with recommended hand hygiene practices pretty directly illustrates the potential consequences of fatigue, in that risks of infection are greater when hands are not properly sanitized. Yes, they could not quantify the consequences in that study, but particularly in this day and age of COVID-19, anything that increases the threat of infection is something that we can all embrace as a clearly negative consequence of decision fatigue! Nonetheless, the present study tries to illustrate that point in a novel way, which I can admit is an advance over prior work. How much of an advance that constitutes is something that the AE will have to determine.

Decision letter (RSOS-201059.R0)

Dear Dr Schnall,

The Editors assigned to your paper RSOS-201059 "Quantifying the Cost of Decision Fatigue: Suboptimal Risk Decisions in Finance" have now received comments from reviewers and would like you to revise the paper in accordance with the reviewer comments and any comments from the Editors. Please note this decision does not guarantee eventual acceptance.

Please submit your revised manuscript and required files (see below) no later than 21 days from today's (ie 12-Aug-2020) date. Note: the ScholarOne system will 'lock' if submission of the revision is attempted 21 or more days after the deadline. If you do not think you will be able to meet this deadline please contact the editorial office immediately.

on behalf of the Associate Editor and Professor Essi Viding (Subject Editor)
 openscience@royalsociety.org

Associate Editor Comments to Author:

Thank you for your patience while we engaged reviewers to assess your manuscript. Happily, we've had an overwhelming response, and received an unusually large number of reports - for which we're grateful to the reviewers who have supplied comments - and can now make a recommendation.

Given the commentary of the reviewers, we would like to see a major revision of your paper to take into account their feedback. Ordinarily, a major revision would only permit you a 3-week deadline; however, given the unusual number of comments, a short extension will be possible if required - please contact the editorial office if so.

Reviewer comments to Author:

Reviewer: 1

Comments to the Author(s)

This paper studies the decisions on credit restructuring applications in different time frames in a commercial bank. Authors use it to test the cost of decision fatigue, with the hypothesis that when decisions are taken along the day, decision fatigue drives to choose the default option. Moreover, the authors quantify the costs that decision fatigue has for the bank.

The research questions are important and well defined according to the literature. The statistical analysis, in general, is adequate and allows to answer the questions. I consider that the paper is very interesting, well written and makes a very nice contribution to the literature. I also consider that there are some points that need to be clarified.

Major points:

- Authors propose that they are the first ones that show that decision fatigue produces bad outcomes. I think that it is not exactly so. It is implicit in the previous studies that choosing more often the "default option" is a bad outcome. Authors argue that it is not the case that the "default option" is a "bad option". Although it is true, it is also true that choosing too often the default option is a bad option. For instance, Persson et al (2019) finish their paper arguing that "From a societal point of view, this is an inefficient and arguably unfair use of medical resources" in their analysis of decision fatigue in the context of surgeons' clinical decision making. Inefficient means here that it is "bad". On the other hand, in this paper the authors are the first who quantify the cost, and I think that this certainly is important and deserves credit. I consider that authors should state better that they are the firsts who quantify the cost of decision fatigue rather than the firsts who show that decision fatigue IS costly.

- My second major concern is with respect to the Analysis in Table 6 regarding Time spent per case. As far as I understand the analysis, the authors argue that they extend the logistic regression 1 adding case length as an additional explanatory variable (they described it in that way in the name of the table). If it is the case, the authors should include the coefficients of all the

independent variables as in Table 1. Moreover, and it seems strange, they merge in the set of explanatory variables in Table 6 the dummy variable 16:00-17:00 and 17:00 or later (Table 1) in the variable 16:00 or later (Table 6). I don't see the reason, I suppose it is only a mistake.

Minor points:

- In Materials and Methods, I would prefer if the country of origin of the bank is included. Maybe it cannot be said because of privacy, if it is the case it should be disclaimed. In Footnote 2 authors describe wages in the sector in the country, I suppose that from those data country can be inferred, so it would be better to show it clearly (or to delete these wages).

- The Results section would be enriched with some descriptive statistics. In particular, given the objectives of the paper and the analysis, approval rates by time framing seem a natural description of the data.

- In Results -> Individual baselines of specific credit officers. I am not sure on what are doing the authors here. As I understand, they are adding control variables to regression 1. However, if those control variables are already included in regression 1, it should be explicitly explain in Table 1. If Table 1 is not controlling by those factors, then the new regression should be included somewhere, in order to check if coefficients in Regression 1 are robust or not. If that regression has not been included, given that its importance is only marginal, I suggest to include it as an Appendix.

- Results -> Time of Day for Decision. Since this paragraph is included after the previous one and it refers to Table 1, I cannot know if "Seniority" is controlled for these results or not. Please clarify.

- When the authors compute the repaid amounts, they assume that this is a lower bound estimation of the cost that fatigue has for the bank. However, typically, when the bank accept a restructuration, te bank assumes some costs (for instance, accepting an extension of payments or similar). Since the restructuration costs are not taken into account in the calculation, the estimated costs are not really a lower bound. The authors should discuss the magnitude of these expenditures for the bank. Is there any estimation of how costly is for the bank to accept a restructuration?

Reviewer: 2

Comments to the Author(s)

See attached file.

Reviewer: 3

Comments to the Author(s)

The article "Quantifying the Cost of Decision Fatigue: Suboptimal Risk Decisions in Finance" investigates if loan officers' decisions to approve loan restructuring are affected by decision fatigue. In addition, the study analyses potential consequences of the decision fatigue. The study uses a dataset from one of the major banks including 26 501 decisions of 50 loan officers made throughout one month. The results are in line with decision fatigue and, furthermore, decision fatigue has negative effect on the bank's revenue. I think that the paper is well-written, the research methods are appropriate and the literature review complete. This review details some concerns I have about the study that I suggest being addressed before a decision about publication can be made.

Major comments:

- Authors define “default option” as the option that requires the least cognitive effort. However, this is not a precise definition of a default option. I do agree with the authors that not approving the application is a reasonable default, however, I would like to see more arguments speaking for it. After all, approving the application may be seen as “safer”, i.e., the restructured loans are more likely to be repaid.

- In the “sample size considerations” authors mention that they have sufficiently big sample (26 501 observations) to detect small-size effects. However, it is not clear whether they took into account the correlation between observations of the same decision-maker. In other words, in simple power calculations it is assumed that the observations are independent of each other, however, given the nature of the dataset it is safe to assume that decisions of the same loan officer will be correlated. If these correlations are relatively big, bigger samples are needed to detect the same effects. Having said that, I do agree that the sample is big enough and the power of conducted tests should not be a problem. While it is tricky to determine sample size for longitudinal data (if the clusters are of roughly equal size the formula for design effect can be used), I would suggest authors to at least specify whether they accounted for the correlation between observations and point out how this can affect the sample size calculations.

- The manuscript would benefit from better description of the dataset and the handling process in the Materials and Methods section. Information that would be of interest is: how are the cases assigned to different employees, is the order of cases random, etc. Authors describe very briefly in the Results section under “Case ordering” that cases are not ordered by loan officers, but I think it would be beneficial to describe it early on in the paper for the readers to understand the results and study design.

- The authors try to tackle the problem of potential confounds. I find the analysis conducted to test whether time spent on the case mattered for the approval decision a valuable robustness check. In addition, authors also try to test if case ordering mattered for the repayment success. If there are specific patterns in the order of considered proposals, they can be the reason for the found time-of-the-day effect, not the decision fatigue. However, in Table 5 the explanatory variable “restructuring approved” may already pick up the variance related to time of the day, as these approvals are time-dependent. Another way to show that the ordering of cases was time-independent would be to test (e.g., in a regression setting) if the characteristics of the applications (i.e., loan amount, company size, credit rating, collections cluster and missed payment) change over time. These results could be kept in an electronic supplementary material.

- Why do the time-variables start from 11am rather than from 09am/10am? While I understand the approach, it would be interesting to see the explanation from the authors. Maybe to make it clearer it would be useful to add a graph that shows the fraction of approved cases throughout the day (starting from 08am)?

- “We also hypothesized that the credit officers’ seniority level or their idiosyncratic approval propensity might affect decisions.” I would suggest to move the hypothesis to methods section, where authors describe the remaining predictions. In addition, it is not entirely clear to me whether the indicator variables were introduced in the logistic regression (it is not shown in Table 1) or were tested separately. Furthermore, given that there were 50 decision-makers and 5 levels of seniority, the conclusions regarding seniority should be made with caution

- I would like to see a discussion on the caveats related to the method used to estimate the impact of decision fatigue on bank’s revenue. For example, authors use variable “restructuring approved” (Table 3) as exogenous variable in the regression estimating the success of the loan repayment. However, the approval depends on the characteristic of the case. In other words, it may be that loan officers indeed approved the applications that wouldn’t have been successful otherwise and rejected the applications that would have been unsuccessful anyway. In that way the approval would increase the likelihood of repayment, but does not necessarily mean that that other applications that were rejected would have been successful if they were approved.

Minor comments:

- a typo in the title: “suboptimal” instead of “supoptimal”
- authors use “credit loan approval” interchangeably with “loan restructuring approval”. In my opinion these two terms describe different things and may be misleading for some readers. Thus, I would suggest being clearer/more explicit in the use of different terms and, for instance, stick with the term “loan restructuring approvals”
- the first paragraph in the section “Time of Day for Decision” would fit better in the Materials and Methods section. It makes it clearer how the work of credit officers looks like, when they take breaks etc. which is of interest in the description of methods to better understand the design of the study.
- In the logistic regression in Table 3 authors model success which is defined as “repayment of overdue amount”. However, some of the companies did not have any amount overdue and authors even use “account not overdue” as one of the explanatory variables. I assume that this is just a matter of description, but it would be good if authors better describe what they mean by “success” in Table 3
- The description of the analysis of the consequences of decision fatigue on bank’s revenue is difficult to follow due to its organization. For instance, authors mention on page 4 that 569 additional loan restructuring applications would be approved but they describe the method used to find that value on page 5 in rows 21-32. This description should be moved to the last paragraph on page 4.
- On page 5/13 in rows 33-37 authors present an alternative calculation for the number of approved applications. However, they do not comment on it or use it in the future calculations. It is not clear what is the purpose of these additional calculations.
- missing reference for Vohs et al. 2008
- references are not in alphabetic order

Reviewer: 4

Comments to the Author(s)

In this article, the authors analysed a data set of credit officers working for a bank and deciding to approve or not loan restructuring requests. This decision involves a trade-off because approving a loan comes with financial loss but increases the chance of avoiding bigger financial loss if the company does not repay, but not approving decreases the likelihood for the company to repay.

Using a very rich data set (30 credit officers working for one month, resulting in 26,501 decisions), the authors show that objective factors (i.e., objective risk attributes) as well as the time of the day (presumably decision fatigue) are influencing credit officers’ decisions. They also compute the financial loss resulting from the time of the day influence on officers’ decisions. These are important findings, potentially showing the real-life consequences of decision fatigue. I however have some concerns that should be addressed as well as suggestions for improvement.

Major

Regarding data availability:

1) I am uncomfortable reviewing an article using a data set that is not available in any way. There is no need to specify the name of the commercial bank, and no need to indicate the name of customers. The problem with data that cannot be shared are manifold, one of them has been illustrated during the covid-19 crisis, with a paper based on a confidential data set regarding the use of hydroxychloroquine to treat patients. Alternatively, perhaps the name of the bank could be mentioned, opening the possibility for someone aiming to replicate the analysis to do so?

Regarding the method:

2) Methods are too concise, and the analyses and simulations are too vague. The analyses that have been used are described in the result section but should also be described more thoroughly, in the methods section, indicating clearly the parameters which were used and spelling out the equations.

3) The effect of seniority and the effect of time of day corrected for objective risk attributes are estimated in different regressions. A linear mixed model can be used to assess all the effects together as well as the potential interactions (e.g., junior officers would be more prone to decision fatigue).

Regarding the results:

4) The main effect (time of day on decision) is supported by a unique analysis. I suggest the author do use different strategies to further support the results. For example, I suggest running 30 logistic regressions (one per credit officer) with the dummy variables for each time of day and the regressors describing the objective risk attributes to then test the parameters at the group level against the null distribution. This would be accounting for individual difference between officers. This could also be done for each day separately if for example the day of the week affects decisions.

5) The data should be shown graphically. More specifically, the requests acceptance rate should be plotted by time slot, with standard error as error bars. Then, the model prediction should be overlaid so that the goodness-of-fit can be evaluated. I also suggest plotting the averaged residuals by time of day of 30 regressions with the objective risk attributes as regressors. This would correct the time series for the objective risk – which is a confound here.

6) Perhaps the time of day could be fitted to the decisions as a parametric regressors (e.g., encoded as 1, 2, 3, etc.) instead of using many dummy regressors? The latter is agnostic regarding the shape of time of day effect on decisions, while the former assume a linear effect.

7) More details about the statistics should be reported (e.g., average of the estimate, uncertainty about the estimates, t-value, degree of freedom) besides the p values. The authors should also report goodness of fit metrics, like the R2 or pseudo-R2 as well as the balanced accuracy of the model.

8) Many tests are used for each dummy variable. Applying a correction for multiple comparisons should be considered.

9) Cognitive fatigue has two impacts on the financial loss, to my understanding:

a) A “negative cost”, i.e. a benefit, corresponding to the difference between restructuring and not restructuring the 40.42% of companies that repaid. If these restructuring requests would have been accepted, this would have been detrimental as they would have paid less than what they actually paid.

b) A cost that corresponds to the product between the increase in the likelihood of repaying given the restructuring approval and the amount that would be repaid. The increase in the likelihood of repaying given the restructuring approval should be corrected for objective risks attributes, which are different between the approved and disapproved requests (because it influences the decisions as reported by the authors). In fact, the improvement is not simply the difference in repayment between the approved and rejected requests as it could be interpreted from lines 6-8 page 5 (“For approved restructuring requests the repayment rate was 52.62%, whereas for rejected it was only 38.71%. Thus, the default decision of rejecting a request was associated with a disadvantageous financial outcome for the bank.”). One could assume that the objective risk difference between the approved and rejected requests may fully explain the difference in the repayment proportion.

The logic behind the analyses page 5 is hard to follow, because it is not explicitly described. Also, to my understanding, not all the information that is presented is necessary to compute the

financial cost of decision fatigue in that context and all the necessary information is not presented. I therefore suggest using analyses based on the aforementioned cost definition.

10) The lack of control group and the lack of information regarding the time and the duration of breaks are limitations that the authors mentioned. They may discuss alternative hypothesis, for example the effect of circadian rhythms, or perhaps a prior estimate of the number of acceptance rate per day. In fact, the data cannot support the idea of decision fatigue. I suggest using "time of day" as much as possible and to offer decision fatigue as a main interpretation.

I also have minor suggestions:

11) I suggest using cognitive fatigue instead of ego-depletion as the latter has been deeply criticised if not invalidated (as mentioned in the discussion), while the former has been used for about a century and less criticised.

12) The main questions, the main metrics that will be used, as well as the predictions are not explicitly described in the introduction.

13) Does the day of the week influence the decisions?

Reviewer: 5

Comments to the Author(s)

The present manuscript describes an interesting study that examines the consequences of decision fatigue on financial decision making, specifically credit loan applications. The authors hypothesize that credit officers will be more likely to approve applications when they are less fatigued (in the morning), but will default to reject more applications when they experience greater fatigue. The most intriguing aspect of this research is the data set that the authors were able to access to conduct their work. Namely, they were given access to a large bank's data for the month of March 2016 in which over 26,000 relevant credit loan applications were processed. Examination of these data revealed a pattern consistent with previous work on decision fatigue, illustrating the predicted trend in greater tendency to approval applications in the morning, and a greater rejection rate in the midday and late afternoon. Perhaps the most critical aspect of the present work is that the authors were able to explore whether these decision fatigue effects actually had deleterious consequences for the bank and its decision makers. By extrapolating from the repayment rate, the authors were able to show that the tendency to reject applications due to decision fatigue cost the bank a considerable amount of money. In this way, the authors tout this study as the first to illustrate the negative downstream consequences for decision quality resulting from fatigue/depletion.

As a researcher actively engaged in work relevant to depletion effects, I read this manuscript with great interest. Certainly, this paper builds nicely on recent efforts that show depletion effects in real-world decision making contexts, such as judges' parole decisions (Danziger et al., 2011) or doctors' prescriptions of antibiotics (Linder et al., 2014). The authors examine the same hypothesis in a new setting, focusing here on the financial decision making of bank employees processing credit loan applications. Consistent with the past work, the authors expect that credit officers will be more likely to take the risk of approving credit loan applications when they are less fatigued, but will follow the default of being conservative and reject credit loan applications more when they are fatigued. The results seem to largely support this hypothesis, though the data leave something to be desired. That is, given the nature of data which the authors have to work with, they were unable to track the fatigue levels of the credit officers as neatly as some past studies have. For instance, in the Danziger et al. work, they were able to track fatigue levels throughout the day by noting when the decision makers (judges) took their breaks during the day. Specifically, they noted that after breaks (such as lunch), there was a temporary restoration of behavior corresponding presumably to reduced levels of fatigue, resulting in a scalloped shape

function throughout the day. In the present research, without access to the specific times at which credit officers took their lunch (or other) breaks during the day, the authors had to simply extrapolate when most people took their lunch breaks to infer the presumed fatigue functions for these decision makers. These limitations are simply unavoidable, given the limitations imposed on those granting them access to the present data set, but they do limit the ability to draw clear conclusions about the role that fatigue presumably plays in the observed function of credit approvals. We can however safely assume that people are at their “best” (or at least are less fatigued) at the beginning of their work day and fatigue tends to increase as the day wears on.

Aside from these limitations in terms of the presumed fatigue levels of the decision makers, the ability to draw conclusions about the financial consequences of these decisions relies heavily on projections based on the banks’ historic data of repayment rate of loans. This seems like a fair way to project the potential financial costs of decision fatigue here, but I will admit that I would have loved to have seen a more nuanced analysis of the applications that did or did not get approved as a function of time of day. I know the authors were not able to parse the data in this way, but it would be very informative to see what parameters of the loan applications get prioritized or ignored when decision makers are or are not fatigued. The authors have access to credit rating, loan amount, seniority level of the credit officer, but from a pragmatic standpoint, I was curious as to the potential for other factors (like SES or race) to infiltrate decisions at this point. That is, the authors conjecture that decision makers opt to follow their defaults more when fatigued, and that the default in this case is to be conservative and reject the application. But in other decision making contexts, we have seen that many other defaults can be operative when people are fatigued or stressed or distracted. People tend to rely on cultural stereotypes more, and thus may reject applications from minority group members (or companies or organizations) when fatigued than when at their full capacity. I know the authors are not able to explore these possibilities, but it is the case that there are other defaults that may be operative, if and when decision makers are feeling fatigued, that may govern their judgments and decisions under these conditions. It may not be that there is a drop in approval rate of any or all applications, but only of certain applications. And this point becomes particularly salient when the authors note that they could not control whether the credit officers look at these applications in a random order or organize the applications in some way (such as doing the easier ones first and saving the more difficult ones for later, or vice versa).

All this said, I think the present work is competently done and provides an intriguing new context in which to study the fundamental question of the (negative) consequences of decision fatigue. What seems to be an emerging pattern in studies like this is that decision fatigue (or depletion) effects appear to be more robust in real world settings, where you have real decision makers doing activities over time that cause fatigue. Thus, these studies have been helpful in reinforcing faith that depletion effects exist in the real world, despite the challenges (and failures to replicate) that researchers have observed in lab studies of this phenomenon. I really like the exploration of these questions in this particular financial decision making context, and sincerely appreciate the value of the present data set (with all its richness and external validity, despite the obvious and unavoidable limitations) to serve as the basis for this investigation. So from that perspective, I am very favorable disposed toward the publication of the present work. Still, though, the question to me is the level of contribution of the present work, over and above past real world demonstrations. The authors tout that this investigation affords the possibility of assessing the actual financial consequences of decision fatigue for the bank (in terms of profits or income). I appreciated their efforts in this regard, but I found that argument a bit hollow. The earlier studies which examined health care professionals’ compliance with recommended hand hygiene practices pretty directly illustrates the potential consequences of fatigue, in that risks of infection are greater when hands are not properly sanitized. Yes, they could not quantify the consequences in that study, but particularly in this day and age of COVID-19, anything that increases the threat of infection is something that we can all embrace as a clearly negative

consequence of decision fatigue! Nonetheless, the present study tries to illustrate that point in a novel way, which I can admit is an advance over prior work. How much of an advance that constitutes is something that the AE will have to determine.

===PREPARING YOUR MANUSCRIPT===

===PREPARING YOUR REVISION IN SCHOLARONE===

- 1) One version identifying all the changes that have been made (for instance, in coloured highlight, in bold text, or tracked changes);
 - 2) A 'clean' version of the new manuscript that incorporates the changes made, but does not highlight them.
 - An individual file of each figure (EPS or print-quality PDF preferred [either format should be produced directly from original creation package], or original software format).
 - An editable file of each table (.doc, .docx, .xls, .xlsx, or .csv).
 - An editable file of all figure and table captions.
- Note: you may upload the figure, table, and caption files in a single Zip folder.
- Any electronic supplementary material (ESM).
 - If you are requesting a discretionary waiver for the article processing charge, the waiver form must be included at this step.
 - If you are providing image files for potential cover images, please upload these at this step, and inform the editorial office you have done so. You must hold the copyright to any image provided.
 - A copy of your point-by-point response to referees and Editors. This will expedite the preparation of your proof.

- Ensure that your data access statement meets the requirements at <https://royalsociety.org/journals/authors/author-guidelines/#data>. You should ensure that you cite the dataset in your reference list. If you have deposited data etc in the Dryad repository, please include both the 'For publication' link and 'For review' link at this stage.
- If you are requesting an article processing charge waiver, you must select the relevant waiver option (if requesting a discretionary waiver, the form should have been uploaded at Step 3 'File upload' above).
- If you have uploaded ESM files, please ensure you follow the guidance at <https://royalsociety.org/journals/authors/author-guidelines/#supplementary-material> to include a suitable title and informative caption. An example of appropriate titling and captioning may be found at https://figshare.com/articles/Table_S2_from_Is_there_a_trade-off_between_peak_performance_and_performance_breadth_across_temperatures_for_aerobic_sc_ope_in_teleost_fishes_/3843624.

Author's Response to Decision Letter for (RSOS-201059.R0)

See Appendix B.

RSOS-201059.R1 (Revision)

Review form: Reviewer 1

Is the manuscript scientifically sound in its present form?

Yes

Are the interpretations and conclusions justified by the results?

Yes

Is the language acceptable?

Yes

Do you have any ethical concerns with this paper?

No

Have you any concerns about statistical analyses in this paper?

No

Recommendation?

Accept as is

Comments to the Author(s)

The authors have addressed adequately all my comments. In the new version, the paper identifies better its contribution to the literature.

Only a minor comment. In my previous review I suggested to include the country of origin of the bank. I understand that it is not included by privacy reason. However, I also made the alternative suggestion of eliminating the wages from the paper, because they allow the identification of the country. The authors have clarified in the paper that original data are not in USD in order to show that the country is not US, but my point was different. The current footnote 3 still includes the monthly salaries identified through a Google search. I wanted to describe that that information allows to identify the country of origin, for instance for a person from that country who knows about the wages in the sector. If the authors want to keep the information confidential they could replace those numbers.

The authors have included now the Gini coefficient of their model. Since the paper will be read by researchers from disciplines different from credit risk management, the authors should include a reference.

Review form: Reviewer 3

Is the manuscript scientifically sound in its present form?

Yes

Are the interpretations and conclusions justified by the results?

Yes

Is the language acceptable?

Yes

Do you have any ethical concerns with this paper?

No

Have you any concerns about statistical analyses in this paper?

No

Recommendation?

Accept with minor revision (please list in comments)

Comments to the Author(s)

As I mentioned before, I believe that this paper makes a valuable contribution to the literature on how decision fatigue may affect decision making. However, I believe that the description of methods used in the paper and the results is at times unclear and could be improved by more detailed description. In particular, I have three minor issues that I think would benefit the paper substantially if they are addressed.

I think that power calculations are important and should not be avoided. Thus, despite having large dataset I would encourage the authors to discuss the power of their statistical analysis. However, as I mentioned earlier, one should be careful in applying formulas that assume independence between observations to longitudinal data.

Furthermore, I think that the analysis of the relationship between seniority of the decision maker and loan restructuring decisions should be revised. Authors use the category "superintendent" as reference category, however, they only have one decision maker in that category. This can inflate the results. Thus, I would suggest to instead use a different reference category (for instance, junior analysts). Furthermore, it is good to note that there were only 30 unique decision makers and five seniority categories, and thus the conclusions regarding seniority should be made with big caution.

Furthermore, it would also be valuable for readers to add a summary table with descriptive statistics of the dataset. In fact, authors attempted to describe the data by adding Figure 1. However, Figure 1 is not mentioned in the text at all. Besides, instead of simulated approval rates for an illustrative application, I think it is better to plot the actual (rather than simulated) approval rates for all applications considered within each time frame. In addition, it would be very informative to add the standard error bars as well as the number of applications considered within each time frame. It is currently unclear how many applications are considered within each time frame. This would give an idea of how the raw data looks like.

Review form: Reviewer 4

Is the manuscript scientifically sound in its present form?

Yes

Are the interpretations and conclusions justified by the results?

Yes

Is the language acceptable?

Yes

Do you have any ethical concerns with this paper?

No

Have you any concerns about statistical analyses in this paper?

Yes

Recommendation?

Major revision is needed (please make suggestions in comments)

Comments to the Author(s)

I thank the authors for their answer to my concerns.

I however think that some of my concerns were not fully addressed, especially regarding the following points.

4) Regarding running a regression per credit officer. Part of my comment was to have some analysis to get a sense of the between credit officer variability, and to check that the effect is also valid when credit officers are considered as a random effect. The idea would be to run 30 regressions and to test the decision fatigue at the group level, that is to test whether the distribution of the 30 individual coefficients is different from zero. Instead of being a fishing expedition, this is a unique statistical test, which I still think would be useful. Moreover, it would be informative to plot the distribution of the coefficients for decision fatigue for the group. This would help the reader to know for example whether some credit officers are highly sensitive to decision fatigue and not others, or whether all of them are slightly sensitive to decision fatigue.

4) Regarding plotting the effect. I still believe that a plot with the whole data overlaying model predictions would be informative and unless I missed something such a figure has not been added to the revised version of the manuscript.

7) Regarding goodness-of-fit metrics. The authors reported the Gini coefficient after having discarded the use of pseudo-R2. What method is used to generate a Gini coefficient (e.g., using the CAP curve, the AUC curve)? This metric also has its caveats. Could the authors also report the balanced accuracy? This is particularly suitable for imbalance data and easy to compute, and I do not see a good reason not to report it. This would help the reader to get a sense of the fit quality.

13) Regarding the day of the week effect, I noticed that another reviewer asked the same question. Reporting some statistics regarding the time of the week would be interesting while not being too costly. Without prior literature, a reasonable assumption is that decision fatigue would be stronger at the end of the week than at the beginning, in the same way that it is higher at the end of the day than at the beginning. Interestingly, weekend could be considered as long breaks and nights as shorter breaks and may even provide more evidence for a fatigue effect (rather than e.g., circadian rhythms).

Decision letter (RSOS-201059.R1)

Dear Dr Schnall

On behalf of the Editors, we are pleased to inform you that your Manuscript RSOS-201059.R1 "Quantifying the Cost of Decision Fatigue: Suboptimal Risk Decisions in Finance" has been accepted for publication in Royal Society Open Science subject to minor revision in accordance with the referees' reports. Please find the referees' comments along with any feedback from the Editors below my signature.

Please submit your revised manuscript and required files (see below) no later than 7 days from today's (ie 23-Feb-2021) date. Note: the ScholarOne system will 'lock' if submission of the revision is attempted 7 or more days after the deadline. If you do not think you will be able to meet this deadline please contact the editorial office immediately.

on behalf of Prof Essi Viding (Subject Editor)
openscience@royalsociety.org

Reviewer comments to Author:
Reviewer: 1

Comments to the Author(s)
The authors have addressed adequately all my comments. In the new version, the paper identifies better its contribution to the literature.

Only a minor comment. In my previous review I suggested to include the country of origin of the bank. I understand that it is not included by privacy reason. However, I also made the alternative suggestion of eliminating the wages from the paper, because they allow the identification of the country. The authors have clarified in the paper that original data are not in USD in order to show that the country is not US, but my point was different. The current footnote 3 still includes the monthly salaries identified through a Google search. I wanted to describe that that information allows to identify the country of origin, for instance for a person from that country who knows about the wages in the sector. If the authors want to keep the information confidential they could replace those numbers.

The authors have included now the Gini coefficient of their model. Since the paper will be read by researchers from disciplines different from credit risk management, the authors should include a reference.

Reviewer: 4
Comments to the Author(s)
I thank the authors for their answer to my concerns.

I however think that some of my concerns were not fully addressed, especially regarding the following points.

4) Regarding running a regression per credit officer. Part of my comment was to have some analysis to get a sense of the between credit officer variability, and to check that the effect is also valid when credit officers are considered as a random effect. The idea would be to run 30 regressions and to test the decision fatigue at the group level, that is to test whether the distribution of the 30 individual coefficients is different from zero. Instead of being a fishing expedition, this is a unique statistical test, which I still think would be useful. Moreover, it would be informative to plot the distribution of the coefficients for decision fatigue for the group. This would help the reader to know for example whether some credit officers are highly sensitive to decision fatigue and not others, or whether all of them are slightly sensitive to decision fatigue.

4) Regarding plotting the effect. I still believe that a plot with the whole data overlaying model predictions would be informative and unless I missed something such a figure has not been added to the revised version of the manuscript.

7) Regarding goodness-of-fit metrics. The authors reported the Gini coefficient after having discarded the use of pseudo-R². What method is used to generate a Gini coefficient (e.g., using the CAP curve, the AUC curve)? This metric also has its caveats. Could the authors also report the balanced accuracy? This is particularly suitable for imbalance data and easy to compute, and I do not see a good reason not to report it. This would help the reader to get a sense of the fit quality.

13) Regarding the day of the week effect, I noticed that another reviewer asked the same question. Reporting some statistics regarding the time of the week would be interesting while not being too costly. Without prior literature, a reasonable assumption is that decision fatigue would be stronger at the end of the week than at the beginning, in the same way that it is higher at the end of the day than at the beginning. Interestingly, weekend could be considered as long breaks and nights as shorter breaks and may even provide more evidence for a fatigue effect (rather than e.g., circadian rhythms).

Reviewer: 3

Comments to the Author(s)

As I mentioned before, I believe that this paper makes a valuable contribution to the literature on how decision fatigue may affect decision making. However, I believe that the description of methods used in the paper and the results is at times unclear and could be improved by more detailed description. In particular, I have three minor issues that I think would benefit the paper substantially if they are addressed.

I think that power calculations are important and should not be avoided. Thus, despite having large dataset I would encourage the authors to discuss the power of their statistical analysis. However, as I mentioned earlier, one should be careful in applying formulas that assume independence between observations to longitudinal data.

Furthermore, I think that the analysis of the relationship between seniority of the decision maker and loan restructuring decisions should be revised. Authors use the category "superintendent" as reference category, however, they only have one decision maker in that category. This can inflate the results. Thus, I would suggest to instead use a different reference category (for instance, junior analysts). Furthermore, it is good to note that there were only 30 unique decision makers and five seniority categories, and thus the conclusions regarding seniority should be made with big caution.

Furthermore, it would also be valuable for readers to add a summary table with descriptive statistics of the dataset. In fact, authors attempted to describe the data by adding Figure 1. However, Figure 1 is not mentioned in the text at all. Besides, instead of simulated approval rates for an illustrative application, I think it is better to plot the actual (rather than simulated) approval rates for all applications considered within each time frame. In addition, it would be very informative to add the standard error bars as well as the number of applications considered within each time frame. It is currently unclear how many applications are considered within each time frame. This would give an idea of how the raw data looks like.

===PREPARING YOUR MANUSCRIPT===

===PREPARING YOUR REVISION IN SCHOLARONE===

Please ensure that you include a summary of your paper at Step 2 'Type, Title, & Abstract'. This should be no more than 100 words to explain to a non-scientific audience the key findings of your

research. This will be included in a weekly highlights email circulated by the Royal Society press office to national UK, international, and scientific news outlets to promote your work.

Author's Response to Decision Letter for (RSOS-201059.R1)

See Appendix C.

Decision letter (RSOS-201059.R2)

Dear Dr Schnall,

It is a pleasure to accept your manuscript entitled "Quantifying the Cost of Decision Fatigue: Suboptimal Risk Decisions in Finance" in its current form for publication in Royal Society Open Science.

on behalf of Prof Essi Viding (Subject Editor)
openscience@royalsociety.org

Appendix A

Referee Report

Royal Society Open Science, RSOS-201059

Quantifying the Cost of Decision Fatigue: Suboptimal Risk Decisions in Finance

Comments to the Authors

Summary The paper studies the effect of time of day on decision behavior. Using decision data from loan officers on credit restructuring applications in a bank, the authors find lower approval rates in the late morning and in the late afternoon. Using this data together with the repayment data of the bank's customers allows to quantify the decision quality and to estimate the costs accompanied with lower approval rates. Knowing the approximate working period of the officers allows the authors to interpret the findings as decision fatigue, namely the effect of continuous decision making on decision outcomes.

The data set is an enormous strength of the paper. Being able to detect decision fatigue in a real-world environment and having an objective measure of decision quality is a great contribution to the literature. Choosing a loan environment is great, as it allows to quantify the monetary value of decisions. The paper is well written and easy to follow.

The comments and suggestions that follow are meant to improve the presentation in some places.

- There seems to be a typo in the title of the paper: 'Supoptimal' should be 'Suboptimal'.
- Your data covers one whole month, namely March 2016. To interpret your point predictions on monetary losses for this month, it would be great to read whether this month is representative for any month, or whether most of such loan decisions occur in March.
- You introduce the paper as being on decision fatigue and define it as the decision involving the least cognitive effort. You then argue that deciding in favor of the default option (rejection of the application) fulfills this criterion. However, later in the paper (p. 3 l. 36) you state that rejections include rejections with counterproposals. Maybe you could add more context to the institutional surrounding to make clear that either counterproposals are rarely observed or do not require high levels of cognitive effort (for me as a layman this sounds like more effort than clicking 'accept').
- on p.3 l. 30 you state that 11% of the decisions are made by other people and you therefore exclude them. It would be nice to know whether these 11% are randomly drawn or whether there is some sort of systematic difference in the applications.
- on p.3 l.39-42 you write about the large power you have by observing 26 000 decisions. However, by only having 30 officers, and using officer-fixed-effects in your regressions, 26 000 is not the right number to refer to. In general, I like that you use the fixed effects for the regressions in order to get rid of the differences between officers. I would also add that to the notes in the tables, as it is not always clear for which regressions you apply it. Similarly, it would also help the reader to depict the number of observations used in each regression in the table itself and not only in the text.
- I would guess that there are also differences in decision behavior (e.g. due to decision fatigue) dependent on the day of the week, not only the timing within a day. The supplementary material you uploaded shows that you have this information. It would be nice to exploit it.
- You write that you do not have data on the exact timing of the lunch break, just a likely time window between 1 and 3 pm. Your main conclusion of the results stems from the timing of the

lunch break (as you logically argue that decision fatigue should arise prior to lunch and prior to going home in the evening). Given your data structure and the assumption that officers have some sort of daily routine, would it be possible to detect the lunch breaks on the individual level more precisely and thereby get even stronger results?

- On p.3 l.10 you describe the trade-off for the bank as 'maximizing the number of loans that are repaid without restructuring while minimizing the risk of a loan default', and some lines above you state that approving a request 'results in a loss relative to adherence to the original payment plan'. To get a full picture of the true (long-term) costs of accepting a request, it is crucial to know the institutional setting in the environment you are observing in detail (from the salaries you describe in footnote 2 I assume that your bank is not located in UK): What exactly happens to the customer if a restructuring application is filed, does it decrease the chances of getting another loan in the future? And if loan is not repaid at all? Does the bank stop to do business with the customer for the rest of his life? Do other banks get to know about the default? Does he have to shut down his company? The reason I believe that a detailed understanding of the incentives is interesting for the reader is not only due to societal costs (which is not the focus of your paper and therefore it is okay to dismiss), but mainly for the long-term costs of the bank you are observing and estimating: A bank that is known for accepting many restructuring applications over time might increase the incentives for other customers to renegotiate existing contracts. If such a mechanism was the case, you would be overestimating the cost of decision fatigue.
- Much of your analysis of the cost estimation of decision fatigue rests on the 596 marginal restructuring requests (p.5 l. 10). As the selection of this subsample is crucial for your results, it should be explained in a more straightforward way. The main 'explanation' seems to be the footnote of Table 2, where you write 'the typical loan was defined as the sample median for loan amounts ...'. This should be explained in more detail.
- p.4 l.8-12 you argue that you did not include the missing payment variable to the regressions because 'it was not statistically significant'. In my view, this is not a valid argument.
- p. 5 l. 8 you state that due to the higher repayment rates of approved ratings (52 %) compared to rejected ratings (38 %) it follows that rejecting leads to 'a disadvantageous financial outcome for the bank'. This claim, however, is only proven in the subsequent paragraphs with more sophisticated techniques than just comparing these two numbers.
- p.6 l.11 you provide a t-test to show that there is a significant difference in decision time between acceptance and rejection. In Table 6, however, this coefficient in the relationship is close to zero and insignificant. Is the officer fixed effect the (main) reason for that (e.g. some officers are always faster and reject more applications in general which could explain this difference in findings)?
- on p. 3 l. 56 you write about the size of a company. It would be nice to know how size is measured (number of employees, revenue, ...)
- In Table 1 your coefficient is labeled 'decision' but should be 'timing of decision' or 'decision time'.
- In Table 4 you put 'changing decision time to a period without decision-fatigue'. A cleaner way to describe the table would be 'changing decision time to a period with less decision-fatigue'
- on p.3 l. 38 you write that refusing a proposal is 'too risky', after explaining in line 9-10 that accepting a proposal increases the risk of no payment at all. Maybe you should use another word in line 38.
- on p.3 l. 40 you cite Cohen (1992). This reference does not appear in the bibliography at the end.

Appendix B

Dear Professor Viding, thank you for the opportunity to revise the paper, and for the valuable feedback. Below we list the changes we made in response to the Reviewers' comments:

Reviewer: 1

This paper studies the decisions on credit restructuring applications in different time frames in a commercial bank. Authors use it to test the cost of decision fatigue, with the hypothesis that when decisions are taken along the day, decision fatigue drives to choose the default option. Moreover, the authors quantify the costs that decision fatigue has for the bank.

The research questions are important and well defined according to the literature. The statistical analysis, in general, is adequate and allows to answer the questions. I consider that the paper is very interesting, well written and makes a very nice contribution to the literature.

Response: We appreciate the Reviewer's positive evaluation of our work.

I also consider that there are some points that need to be clarified.

Major points:

- Authors propose that they are the first ones that show that decision fatigue produces bad outcomes. I think that it is not exactly so. It is implicit in the previous studies that choosing more often the "default option" is a bad outcome. Authors argue that it is not the case that the "default option" is a "bad option". Although it is true, it is also true that choosing too often the default option is a bad option. For instance, Persson et al (2019) finish their paper arguing that "From a societal point of view, this is an inefficient and arguably unfair use of medical resources" in their analysis of decision fatigue in the context of surgeons' clinical decision making. Inefficient means here that it is "bad". On the other hand, in this paper the author are the first who quantify the cost, and I think that this certainly is important and deserves credit. I consider that authors should state better that they are the firsts who quantify the cost of decision fatigue rather than the firsts who show that decision fatigue IS costly.

Response: Thank you for pointing out that the novelty of our work involves quantifying the cost of bad outcomes resulting from decision fatigue, rather than in documenting that such decisions are indeed bad. We have now clarified this in the Summary, on p. 1 and on p. 5.

- My second major concern is with respect to the Analysis in Table 6 regarding Time spent per case. As far as I understand the analysis, the authors argue that they extend the logistic regression 1 adding case length as an additional explanatory variable (they described it in that way in the name of the table). If it is the case, the authors should include the coefficients of all the independent variables as in Table 1. Moreover, and it seems strange, they merge in the set of explanatory variables in Table 6 the dummy variable 16:00-17:00 and 17:00 or later (Table 1) in the variable 16:00 or later (Table 6). I don't see the reason, I suppose it is only a mistake.

Thanks for catching this, it was indeed an oversight. We now revised Table 6 to accurately reflect this information.

Minor points:

- In Materials and Methods, I would prefer if the country of origin of the bank is included. Maybe it cannot be said because of privacy, if it is the case it should be disclaimed. In Footnote 2 authors describe wages in the sector in the country, I suppose that from those data country can be inferred, so it would be better to show it clearly (or to delete these wages).

Response: Unfortunately given the non-disclosure agreement we cannot share additional information about the bank, because it is one of the largest banks in that country. In Footnote 2 we use dollar amounts to describe salaries because USD is a universally understood currency. It is, however, not the currency of the country, which we have now clarified in the Footnote.

- The Results section would be enriched with some descriptive statistics. In particular, given the objectives of the paper and the analysis, approval rates by time framing seem a natural description of the data.

Thank you for the suggestion, we now added a Figure to illustrate the approval rates more directly.

- In Results -> Individual baselines of specific credit officers. I am not sure on what are doing the authors here. As I understand, they are adding control variables to regression 1. However, if those control variables are already included in regression 1, it should be explicitly explain in Table 1. If Table 1 is not controlling by those factors, then the new regression should be included somewhere, in order to check if coefficients in Regression 1 are robust or not. If that regression has not been included, given that its importance is only marginal, I suggest to include it as an Appendix.

Thanks for this suggestion, we now added this information to Table 1.

- Results -> Time of Day for Decision. Since this paragraph is included after the previous one and it refers to Table 1, I cannot know if "Seniority" is controlled for these results or not. Please clarify.

We now included this information in Table 1.

- When the authors compute the repaid amounts, they assume that this is a lower bound estimation of the cost that fatigue has for the bank. However, typically, when the bank accept a restructuring, the bank assumes some costs (for instance, accepting an extension of payments or similar). Since the restructuring costs are not taken into account in the calculation, the estimated costs are not really a lower bound. The authors should discuss the magnitude of these expenditures for the bank. Is there any estimation of how costly is for the bank to accept a restructuring?

Response: If a loan is restructured and paid by the customer, the operating cost for the bank is very low because the loan repayment is automatically collected by the bank's IT system from the customer's account. By contrast, if the recovery fails, the bank is bound to incur high operating cost – e.g., there will be numerous attempts by the bank's staff to contact the customer as well as potential cost for legal action and repossession of physical collateral. This means that if we had included such cost, the cost of fatigue would have been even higher. By ignoring such cost we both paint a conservative picture (underestimating the cost of fatigue) and avoid having to make further assumptions – as the data was collected quite soon after the actual decision was taken whereas the collection cost is incurred over a period of many years (collection efforts typically spread out over up to 7 years), the actual collection cost was not known yet and we would have had to estimate what cost the bank may incur in the future.

Reviewer: 2

The paper studies the effect of time of day on decision behavior. Using decision data from loan officers on credit restructuring applications in a bank, the authors find lower approval rates in the late morning and in the late afternoon. Using this data together with the repayment data of the bank's customers allows to quantify the decision quality and to estimate the costs accompanied with lower approval rates. Knowing the approximate working period of the officers allows the authors to interpret the findings as decision fatigue, namely the effect of continuous decision making on decision outcomes.

The data set is an enormous strength of the paper. Being able to detect decision fatigue in a real-world environment and having an objective measure of decision quality is a great contribution to the literature.

Choosing a loan environment is great, as it allows to quantify the monetary value of decisions. The paper is well written and easy to follow. The comments and suggestions that follow are meant to improve the presentation in some places.

Response: We appreciate the Reviewer's positive evaluation of our work.

- There seems to be a typo in the title of the paper: 'Supoptimal' should be 'Suboptimal'.

Response: Thanks for catching this typo, which we now fixed.

- Your data covers one whole month, namely March 2016. To interpret your point predictions on monetary losses for this month, it would be great to read whether this month is representative for any month, or whether most of such loan decisions occur in March.

Response: Decisions for March are indeed relatively typical for the number of proposals that are processed across the year.

- You introduce the paper as being on decision fatigue and define it as the decision involving the least cognitive effort. You then argue that deciding in favor of the default option (rejection of the application) fulfills this criterion. However, later in the paper (p. 3 l. 36) you state that rejections include rejections with counterproposals. Maybe you could add more context to the institutional surrounding to make clear that either counterproposals are rarely observed or do not require high levels of cognitive effort (for me as a layman this sounds like more effort than clicking 'accept').

Response: Thanks for drawing our attention to this definitional issue. Technically default decisions do not necessarily always involve the least effort, but they involve relatively little effort. We have now clarified this in the summary and on p. 1 (last paragraph).

Regarding counterproposals, rejections included outright rejections or rejections combined with a counterproposal because a distinction between the two forms of rejection was not possible due to a limitation of the bank's information technology system, which we note on p. 2.

- on p.3 l. 30 you state that 11% of the decisions are made by other people and you therefore exclude them. It would be nice to know whether these 11% are randomly drawn or whether there is some sort of systematic difference in the applications.

Response: As noted, the other credit officers who helped out occasionally likely differed in various ways from the full-time staff, and moreover, they also worked on other tasks beyond credit approvals, which might contribute to decision fatigue to a different extent. Since we had no other way of controlling for those confounds, we excluded their data.

- on p.3 l.39-42 you write about the large power you have by observing 26 000 decisions. However, by only having 30 officers, and using officer-fixed-effects in your regressions, 26 000 is not the right number to refer to. In general, I like that you use the fixed effects for the regressions in order to get rid of the differences between officers. I would also add that to the notes in the tables, as it is not always clear for which regressions you apply it. Similarly, it would also help the reader to depict the number of observations used in each regression in the table itself and not only in the text.

Response: We agree with the reviewer that given the complexity of the analysis, and the fact that each credit officer contributes data over the course of an entire month, it would be too simplistic to make such a generic statement about the sample size. Indeed, Reviewer 3 also raised this point, and overall all reviewers stated being impressed that we had access to such a rich data set. We therefore omitted the statement referring to

sample, since, as other reviewers also noted, it is self-evident that the sample was very large and therefore sufficiently powered.

- I would guess that there are also differences in decision behavior (e.g. due to decision fatigue) dependent on the day of the week, not only the timing within a day. The supplementary material you uploaded shows that you have this information. It would be nice to exploit it.

Response: We are not aware of any literature that would provide a rationale for investigating effects of day of the week, and therefore were unable to derive any predictions. This would be a different research question that goes beyond the focus of the paper.

- You write that you do not have data on the exact timing of the lunch break, just a likely time window between 1 and 3 pm. Your main conclusion of the results stems from the timing of the lunch break (as you logically argue that decision fatigue should arise prior to lunch and prior to going home in the evening). Given your data structure and the assumption that officers have some sort of daily routine, would it be possible to detect the lunch breaks on the individual level more precisely and thereby get even stronger results?

Response: We agree that having more information on the precise lunch breaks would be ideal. Unfortunately, as noted, given flexible work patterns we were unable to analyse this. However, we consider the applied relevance of our findings to be a major benefit: By being aware of the decision fatigue that we documented, banks and other businesses can put in place measures to mitigate such effects, without necessarily having to specify whether each individual is affected equally. Thus, despite this limitation we believe that the results can be impactful in the real-world.

- On p.3 l.10 you describe the trade-off for the bank as 'maximizing the number of loans that are repaid without restructuring while minimizing the risk of a loan default', and some lines above you state that approving a request 'results in a loss relative to adherence to the original payment plan'. To get a full picture of the true (long-term) costs of accepting a request, it is crucial to know the institutional setting in the environment you are observing in detail (from the salaries you describe in footnote 2 I assume that your bank is not located in UK): What exactly happens to the customer if a restructuring application is filed, does it decrease the chances of getting another loan in the future? And if loan is not repaid at all? Does the bank stop to do business with the customer for the rest of his life? Do other banks get to know about the default? Does he have to shut down his company? The reason I believe that a detailed understanding of the incentives is interesting for the reader is not only due to societal costs (which is not the focus of your paper and therefore it is okay to dismiss), but mainly for the long-term costs of the bank you are observing and estimating: A bank that is known for accepting many restructuring applications over time might increase the incentives for other customers to renegotiate existing contracts. If such a mechanism was the case, you would be overestimating the cost of decision fatigue.

Response: Thank you asking for further detail on the bank's decision making process, since this is likely something other academic readers are also unfamiliar with. We now added the following Footnote 1: Both a restructuring and a failure to repay a loan at all are defined as a default by the central bank and therefore recorded as a breach of contract by the country's credit bureaus – this information then is shared with other banks. If a customer restructures, he or she agrees to a new repayment schedule; if this schedule is kept, banks generally maintain the relationship with the customer and after a while (typically 1-2 years) will even consider new loans to the customer. If a customer fails to repay the loan, banks will terminate all lending business with the customer and typically not consider new loans to the customer for many years.

- Much of your analysis of the cost estimation of decision fatigue rests on the 596 marginal restructuring requests (p.5 l. 10). As the selection of this subsample is crucial for your results, it should be explained in a more straightforward way. The main 'explanation' seems to be the footnote of Table 2, where you write 'the typical loan was defined as the sample median for loan amounts ...'. This should be explained in more detail.

We already include the following sentence in the main text: " We then estimated the incrementally repaid amount by multiplying each loan's increased repayment probability with the loan amount, adding up the incremental expectation in recovery for each of the 569 loans." If the editor thinks this needs more detail we would be glad to add it.

- p.4 l.8-12 you argue that you did not include the missing payment variable to the regressions because 'it was not statistically significant'. In my view, this is not a valid argument.

"Account not overdue" now captures both the 1421 cases where the customer never had missed a payment and the 1469 cases where the customer had missed a payment in the past but at the time when the restructuring was requested, the account was not overdue. The additional dummy variable for a missed payment would only have further differentiated these two very similar sub-groups. Adding insignificant variables to a regression renders it mis-specified and increases the risk of multicollinearity, especially if related variables are included (in our case, the collections cluster and the flag whether the account is currently overdue). As it was not our research objective to determine whether or not missed payments change the outcome but only to control for effects other than fatigue also impacting outcomes, it appeared prudent to us to prune the model from irrelevant factors. This is a standard statistical technique.

- p. 5 l. 8 you state that due to the higher repayment rates of approved ratings (52 %) compared to rejected ratings (38 %) it follows that rejecting leads to 'a disadvantageous financial outcome for the bank'. This claim, however, is only proven in the subsequent paragraphs with more sophisticated techniques than just comparing these two numbers.

Response: This is a good point. We now rewrote the sentence to say: "As a next step we explored whether the default decision of rejecting a request was associated with a disadvantageous financial outcome for the bank." to set up the following paragraph.

- p.6 l.11 you provide a t-test to show that there is a significant difference in decision time between acceptance and rejection. In Table 6, however, this coefficient in the relationship is close to zero and insignificant. Is the officer fixed effect the (main) reason for that (e.g. some officers are always faster and reject more applications in general which could explain this difference in findings)?

Indeed, the results from Table 6 suggest that the other variables are better predictors of approval. When we run a regression to explain case length, we do find that the fixed effects are highly significant – i.e., your hunch is correct that some officers are always faster and reject more applications in general.

- on p. 3 l. 56 you write about the size of a company. It would be nice to know how size is measured (number of employees, revenue, ...)

Size is an attribute assigned by the bank. It is loosely correlated with turnover but also reflects operational considerations (e.g., a company that starts out in one segment is often kept there even if it grows or shrinks to a size outside of that segment in order to ensure continuity with the relationship manager; in certain situations, a company also might be assigned to a segment by other criteria such as total assets or lending volume).

- In Table 1 your coefficient is labeled 'decision' but should be 'timing of decision' or 'decision time'.

Response: This is a good point, so we replaced the label with 'decision time'.

- In Table 4 you put 'changing decision time to a period without decision-fatigue'. A cleaner way to describe the table would be 'changing decision time to a period with less decision-fatigue'

Response: This is also a good point, so we changed the table heading.

- on p.3 l. 38 you write that refusing a proposal is 'too risky', after explaining in line 9-10 that accepting a proposal increases the risk of no payment at all. Maybe you should use another word in line 38.

Response: This is a good point, so reworded the sentence to read: "Both types of rejections, however, captured the same construct, namely that the credit officer deemed the probably of the client repaying the loan to be very low.

- on p.3 l. 40 you cite Cohen (1992). This reference does not appear in the bibliography at the end.

Response: We no longer refer to sample size, and therefore the reference is no longer needed.

Reviewer: 3

The article "Quantifying the Cost of Decision Fatigue: Suboptimal Risk Decisions in Finance" investigates if loan officers' decisions to approve loan restructuring are affected by decision fatigue. In addition, the study analyses potential consequences of the decision fatigue. The study uses a dataset from one of the major banks including 26 501 decisions of 50 loan officers made throughout one month. The results are in line with decision fatigue and, furthermore, decision fatigue has negative effect on the bank's revenue. I think that the paper is well-written, the research methods are appropriate and the literature review complete. This review details some concerns I have about the study that I suggest being addressed before a decision about publication can be made.

Major comments:

- Authors define "default option" as the option that requires the least cognitive effort. However, this is not a precise definition of a default option. I do agree with the authors that not approving the application is a reasonable default, however, I would like to see more arguments speaking for it. After all, approving the application may be seen as "safer", i.e., the restructured loans are more likely to be repaid.

Response: Reviewer 2 also raised this point so we are now more clear about how the default option is defined, so we fixed this to say that the default involves relatively little effort. In the particular context of the current research, we empirically showed that first, rejecting the application was the default, and second that approving the application was safer, but this is not something that could necessarily be expected a priori. Thus, part of the objective was to determine what the default decision was, and whether it had positive or negative outcomes for the bank.

- In the "sample size considerations" authors mention that they have sufficiently big sample (26 501 observations) to detect small-size effects. However, it is not clear whether they took into account the correlation between observations of the same decision-maker. In other words, in simple power calculations it is assumed that the observations are independent of each other, however, given the nature of the dataset it is safe to assume that decisions of the same loan officer will be correlated. If these correlations are relatively big, bigger samples are needed to detect the same effects. Having said that, I do agree that the sample is big enough and the power of conducted tests should not be a problem. While it is tricky to determine sample size for longitudinal data (if the clusters are of roughly equal size the formula for design effect can be used), I would suggest authors to at least specify whether they accounted for the correlation between observations and point out how this can affect the sample size calculations.

Response: The Reviewer picks up on a point that was also raised by Reviewer 2, namely that given the complexity of the longitudinal data set it would be too simplistic to describe the sample size in conventional terms given Cohen's (1992) classification. Given that several reviewers noted that the large sample as a major benefit of the research, we omitted further discussion of this aspect that is relatively self-evident.

- The manuscript would benefit from better description of the dataset and the handling process in the Materials and Methods section. Information that would be of interest is: how are the cases assigned to different employees, is the order of cases random, etc. Authors describe very briefly in the Results section under “Case ordering” that cases are not ordered by loan officers, but I think it would be beneficial to describe it early on in the paper for the readers to understand the results and study design.

Response: There are multiple work queues that are processed in sequential order – small/medium/large cases are randomly assigned amongst all credit officers with junior/medium/senior seniority. Excess volume (load balances) is picked up by other credit officers in sequential order; cases referred to more senior credit officers due to complexity are also actioned sequentially. Therefore all work queues amount to a random assignment. We have now added this information on p. 5 in the section of Case ordering.

- The authors try to tackle the problem of potential confounds. I find the analysis conducted to test whether time spent on the case mattered for the approval decision a valuable robustness check. In addition, authors also try to test if case ordering mattered for the repayment success. If there are specific patterns in the order of considered proposals, they can be the reason for the found time-of-the-day effect, not the decision fatigue. However, in Table 5 the explanatory variable “restructuring approved” may already pick up the variance related to time of the day, as these approvals are time-dependent. Another way to show that the ordering of cases was time-independent would be to test (e.g., in a regression setting) if the characteristics of the applications (i.e., loan amount, company size, credit rating, collections cluster and missed payment) change over time. These results could be kept in an electronic supplementary material.

Response: Given that the applications are randomly allocated to credit officers, and given the high number of applications processed, it is not plausible that characteristics of applications would change over time. We therefore would not consider this an informative analysis, and instead, believe it would be confusing to readers.

- Why do the time-variables start from 11am rather than from 09am/10am? While I understand the approach, it would be interesting to see the explanation from the authors. Maybe to make it clearer it would be useful to add a graph that shows the fraction of approved cases throughout the day (starting from 08am)?

Response: As noted, credit officers follow relatively flexible work patterns, and therefore we could consider it inappropriate to conduct analyses that leave out this important confound.

- “We also hypothesized that the credit officers’ seniority level or their idiosyncratic approval propensity might affect decisions.” I would suggest to move the hypothesis to methods section, where authors describe the remaining predictions. In addition, it is not entirely clear to me whether the indicator variables were introduced in the logistic regression (it is not shown in Table 1) or were tested separately. Furthermore, given that there were 50 decision-makers and 5 levels of seniority, the conclusions regarding seniority should be made with caution

We now added this information to Table 1.

- I would like to see a discussion on the caveats related to the method used to estimate the impact of decision fatigue on bank’s revenue. For example, authors use variable “restructuring approved” (Table 3) as exogenous variable in the regression estimating the success of the loan repayment. However, the approval depends on the characteristic of the case. In other words, it may be that loan officers indeed approved the applications that wouldn’t have been successful otherwise and rejected the applications that would have been unsuccessful anyway. In that way the approval would increase the likelihood of repayment, but does not necessarily mean that that other applications that were rejected would have been successful if they were approved.

For this reason we have controlled for all the known loan characteristics in modelling the probability of loan repayment. It is true, however, that we don't know what would have happened in an alternative context where these rejected loan restructurings had been approved – therefore we only can present an expected impact, and consistent with us operating with probabilities (as opposed to certain outcomes), we are only capturing the expected increase in the probability of repayment – e.g., if we estimate the probability of repayment to rise from 20% to 21%, we only capture that 1% of the loan amount as potential impact.

Minor comments:

- a typo in the title: “suboptimal” instead of “supoptimal”

Response: Thanks for catching this typo, which we now fixed.

- authors use “credit loan approval” interchangeably with “loan restructuring approval”. In my opinion these two terms describe different things and may be misleading for some readers. Thus, I would suggest being clearer/more explicit in the use of different terms and, for instance, stick with the term “loan restructuring approvals”

Response: We considered this comment and debated what kind of language would be best to convey the key constructs. While we could see the benefits of using the same term throughout, when trying it, some parts of the text became repetitive. We therefore opted to keep the current language, but would be glad to change it if the Editor thinks it would be beneficial.

- the first paragraph in the section “Time of Day for Decision” would fit better in the Materials and Methods section. It makes it clearer how the work of credit officers looks like, when they take breaks etc. which is of interest in the description of methods to better understand the design of the study.

Response: While we appreciate the reviewer’s suggestion, we think that this paragraph fits better in the Results section, because it describes how we processed the data for analyses. But if the editor would prefer to have this section moved, we would be glad to do so.

- In the logistic regression in Table 3 authors model success which is defined as “repayment of overdue amount”. However, some of the companies did not have any amount overdue and authors even use “account not overdue” as one of the explanatory variables. I assume that this is just a matter of description, but it would be good if authors better describe what they mean by “success” in Table 3

There are situations where the bank becomes aware of problems on the borrower's side before the borrower has become late on any payment – e.g., the borrower proactively approaches the bank or the bank reads in the local newspaper that a company has initiated bankruptcy proceedings. These accounts are also given to the collections area of the bank that customarily refers to the amount owed as the "overdue amount" even if in these situations, the term "overdue" is technically not yet correct. "Success" is that the bank considers the account back in good order which includes an objective element that the account is not overdue and a subjective element that the bank is not aware of any other reason why the account is unlikely to be collected. To make this more clear, we use the expression of “successful repayment of the overdue amount.”

- The description of the analysis of the consequences of decision fatigue on bank’s revenue is difficult to follow due to its organization. For instance, authors mention on page 4 that 569 additional loan restructuring applications would be approved but they describe the method used to find that value on page 5 in rows 21-32. This description should be moved to the last paragraph on page 4.

Response: The two parts of the results capture two different processes, namely arriving at the number of additional loan proposals, and then the cost of non-repayment of those loans. For the former we also provide an alternative way of determining this number, which we consider important as well. We believe

that this order of presenting the results most efficiently outlines the sequence of steps involved.

- On page 5/13 in rows 33-37 authors present an alternative calculation for the number of approved applications. However, they do not comment on it or use it in the future calculations. It is not clear what is the purpose of these additional calculations.

As we are estimating the impact (i.e., we calculate outcomes for an alternative, hypothetical situation), we felt that estimating the impact with a second, different methodology would provide a benchmark to judge the more detailed estimation. The fact that both methodologies yield very similar results gives us additional confidence that our estimates are reasonable.

- missing reference for Vohs et al. 2008

Response: We now fixed this.

- references are not in alphabetic order

Response: We also fixed this.

Reviewer: 4

In this article, the authors analysed a data set of credit officers working for a bank and deciding to approve or not loan restructuring requests. This decision involves a trade-off because approving a loan comes with financial loss but increases the chance of avoiding bigger financial loss if the company does not repay, but not approving decreases the likelihood for the company to repay. Using a very rich data set (30 credit officers working for one month, resulting in 26,501 decisions), the authors show that objective factors (i.e., objective risk attributes) as well as the time of the day (presumably decision fatigue) are influencing credit officers' decisions. They also compute the financial loss resulting from the time of the day influence on officers' decisions. These are important findings, potentially showing the real-life consequences of decision fatigue.

Response: We appreciate the Reviewer's positive evaluation of our work.

I however have some concerns that should be addressed as well as suggestions for improvement.

Major

Regarding data availability:

1) I am uncomfortable reviewing an article using a data set that is not available in any way. There is no need to specify the name of the commercial bank, and no need to indicate the name of customers. The problem with data that cannot be shared are manifold, one of them has been illustrated during the covid-19 crisis, with a paper based on a confidential data set regarding the use of hydroxychloroquine to treat patients. Alternatively, perhaps the name of the bank could be mentioned, opening the possibility for someone aiming to replicate the analysis to do so?

Response: The reviewer is correct that it is certainly suboptimal to not be able to share the data. We spent considerable time trying to solve this, but were unable to get around the issue. Ultimately it is a judgment call regarding whether it would be better to publicly share the findings in the hope that they will be informative, or keep them in a file drawer with no prospect of them ever being of any benefit to anyone. We believe the former to be the better course of action, and so we consulted the journal's editors. Based on their input, we arrived at the current data sharing agreement.

As a side note, we do not fully see the relevance of a hydroxycloquine study to our research. Any individual study can have problems due to many reasons. We do not think it is possible to infer that one problematic study makes other studies also problematic when the only common factor is whether data were publicly shared. By the same token, even when data are available, they may never be independently inspected or verified, so it is not necessarily a guarantee that findings are more reliable.

Regarding the method:

2) Methods are too concise, and the analyses and simulations are too vague. The analyses that have been used are described in the result section but should also be described more thoroughly, in the methods section, indicating clearly the parameters which were used and spelling out the equations.

Response: Based on the specific comments from the other reviewers we elaborated on both methods and results, and believe they are now more thorough and more clear.

3) The effect of seniority and the effect of time of day corrected for objective risk attributes are estimated in different regressions. A linear mixed model can be used to assess all the effects together as well as the potential interactions (e.g., junior officers would be more prone to decision fatigue).

Response: We had modelled the fixed effects in the same regression and only omitted the 29 fixed effect variables in Table 1 because we felt showing these additional 29 lines of coefficients would be a poor use of the journal's space. As we are modelling a binary outcome, we cannot use a linear model and instead need a logistic model. We attempted a mixed effect logistic regression but because of multicollinearity amongst the many interaction terms, this model did not converge.

Regarding the results:

4) The main effect (time of day on decision) is supported by a unique analysis. I suggest the author do use different strategies to further support the results. For example, I suggest running 30 logistic regressions (one per credit officer) with the dummy variables for each time of day and the regressors describing the objective risk attributes to then test the parameters at the group level against the null distribution. This would be accounting for individual difference between officers. This could also be done for each day separately if for example the day of the week affects decisions.

Response: While we appreciate that running separate analyses for each participant might seem intuitive, we worry that running 30 x 7 (credit officers x day of the week) separate analysis would vastly increase the probability of observing a spurious effect. Indeed, some readers might misinterpret such an approach as a "fishing expedition," especially if we include day of the week, for which we had no prediction given that there is no relevant literature suggesting such an effect. We therefore are uncomfortable with running all these analyses, especially given the journal's focus on open science, but would be glad to consider doing so if the Editor considers it essential.

5) The data should be shown graphically. More specifically, the requests acceptance rate should be plotted by time slot, with standard error as error bars. Then, the model prediction should be overlaid so that the goodness-of-fit can be evaluated. I also suggest plotting the averaged residuals by time of day of 30 regressions with the objective risk attributes as regressors. This would correct the time series for the objective risk – which is a confound here.

Response: We now plot the data, as also suggested by Reviewer 1. However, as noted above, do not consider it appropriate to describe 30 separate regressions, and indeed, believe this would be confusing to readers.

6) Perhaps the time of day could be fitted to the decisions as a parametric regressors (e.g., encoded as 1, 2, 3,

etc.) instead of using many dummy regressors? The latter is agnostic regarding the shape of time of day effect on decisions, while the former assume a linear effect.

Response: The time of day is not ordinal because we don't know exactly when people go for lunch. We expect that people go out for lunch at different times and that most people take out a break of 30 to 90 minutes some time between noon and 15:00. The results confirm this, there is an ordinal downward trend from morning until noon, then an upward trend until 15:00, and then again a downward trend. Hence dummy variables are the only way to capture this.

7) More details about the statistics should be reported (e.g., average of the estimate, uncertainty about the estimates, t-value, degree of freedom) besides the p values. The authors should also report goodness of fit metrics, like the R2 or pseudo-R2 as well as the balanced accuracy of the model.

Standard Errors (SE) are the standard way of quantifying uncertainty of estimates and are indicated in our output. t-values are not applicable to logistic regression, instead the z-statistic (Wald) and p-values are used to indicate the significance level of a relationship. As z-statistics and p-values express the same information and readers are more familiar with the latter, we have only given the p-values. Also R2 is not applicable to logistic regressions; pseudo-R2 has various limitations and a better metric is the Gini coefficient which we are happy to add as a metric of goodness of fit. The model has a Gini coefficient of 0.448 which is a good fit and comparable to the goodness of fit of the credit rating models that banks typically use in this kind of portfolio. There are 26451 residual degrees of freedom. We now have added this information in the Methods section on p. 3.

8) Many tests are used for each dummy variable. Applying a correction for multiple comparisons should be considered.

Response: While we understand that this may be an issue with smaller data sets we believe that with 26451 residual degrees of freedom this is not necessary.

9) Cognitive fatigue has two impacts on the financial loss, to my understanding:

a) A “negative cost”, i.e. a benefit, corresponding to the difference between restructuring and not restructuring the 40.42% of companies that repaid. If these restructuring requests would have been accepted, this would have been detrimental as they would have paid less than what they actually paid.

b) A cost that corresponds to the product between the increase in the likelihood of repaying given the restructuring approval and the amount that would be repaid. The increase in the likelihood of repaying given the restructuring approval should be corrected for objective risks attributes, which are different between the approved and disapproved requests (because it influences the decisions as reported by the authors). In fact, the improvement is not simply the difference in repayment between the approved and rejected requests as it could be interpreted from lines 6-8 page 5 (“For approved restructuring requests the repayment rate was 52.62%, whereas for rejected it was only 38.71%. Thus, the default decision of rejecting a request was associated with a disadvantageous financial outcome for the bank.”). One could assume that the objective risk difference between the approved and rejected requests may fully explain the difference in the repayment proportion.

Response: Our impact calculation does explicitly consider the objective risk attributes and captures only the marginal impact of the restructuring, as described in the paragraph beginning with " In order to estimate the incremental number of loans that would have been repaid..." The probability of repayment considers all loan attributes such as the amount, risk rating, collections cluster, etc., so we believe the relevant information is already captured. But if the editor feels that additional information is needed, we would be glad to add it.

The logic behind the analyses page 5 is hard to follow, because it is not explicitly described. Also, to my understanding, not all the information that is presented is necessary to compute the financial cost of decision fatigue in that context and all the necessary information is not presented. I therefore suggest using analyses based on the aforementioned cost definition.

Response: As the reviewer says, it is important to consider the objective risk attributes of each loan, and therefore our approach is necessary to do the same. Based on the other Reviewers' suggestions we already added some additional information to various analyses, but we would be glad to do more if the editor recommends it.

10) The lack of control group and the lack of information regarding the time and the duration of breaks are limitations that the authors mentioned. They may discuss alternative hypothesis, for example the effect of circadian rhythms, or perhaps a prior estimate of the number of acceptance rate per day. In fact, the data cannot support the idea of decision fatigue. I suggest using "time of day" as much as possible and to offer decision fatigue as a main interpretation.

Response: This is a good point so we have clarified that our results are consistent with decision fatigue rather than it definitely being the cause behind the findings.

I also have minor suggestions:

11) I suggest using cognitive fatigue instead of ego-depletion as the latter has been deeply criticised if not invalidated (as mentioned in the discussion), while the former has been used for about a century and less criticised.

Response: While we do not necessarily agree that ego-depletion has been "invalidated", it is indeed a complex topic that goes beyond the focus of the present paper. We therefore now provide a more nuanced discussion in the paper, and as noted above make clear that our effects are consistent with decision fatigue.

12) The main questions, the main metrics that will be used, as well as the predictions are not explicitly described in the introduction.

Response: We are not entirely sure what specific aspects the Reviewer is referring to. We hope that our revisions in response to all Reviewers have clarified these issues, but would be glad to take further guidance from the Editor if more needs to be done.

13) Does the day of the week influence the decisions?

Response: We did not have any predictions for differences as a function of day of the week given that there is no existing literature to suggest this, and therefore did not test this question.

Reviewer: 5

The present manuscript describes an interesting study that examines the consequences of decision fatigue on financial decision making, specifically credit loan applications. The authors hypothesize that credit officers will be more likely to approve applications when they are less fatigued (in the morning), but will default to reject more applications when they experience greater fatigue. The most intriguing aspect of this research is the data set that the authors were able to access to conduct their work. Namely, they were given access to a large bank's data for the month of March 2016 in which over 26,000 relevant credit loan applications were processed. Examination of these data revealed a pattern consistent with previous work on decision fatigue, illustrating the predicted trend in greater tendency to approval applications in the morning, and a greater rejection rate in the midday and late afternoon. Perhaps the most critical aspect of the present work is that the authors were able to explore whether these decision fatigue effects actually had deleterious consequences

for the bank and its decision makers. By extrapolating from the repayment rate, the authors were able to show that the tendency to reject applications due to decision fatigue cost the bank a considerable amount of money. In this way, the authors tout this study as the first to illustrate the negative downstream consequences for decision quality resulting from fatigue/depletion.

As a researcher actively engaged in work relevant to depletion effects, I read this manuscript with great interest. Certainly, this paper builds nicely on recent efforts that show depletion effects in real-world decision making contexts, such as judges' parole decisions (Danziger et al., 2011) or doctors' prescriptions of antibiotics (Linder et al., 2014). The authors examine the same hypothesis in a new setting, focusing here on the financial decision making of bank employees processing credit loan applications. Consistent with the past work, the authors expect that credit officers will be more likely to take the risk of approving credit loan applications when they are less fatigued, but will follow the default of being conservative and reject credit loan applications more when they are fatigued.

The results seem to largely support this hypothesis, though the data leave something to be desired. That is, given the nature of data which the authors have to work with, they were unable to track the fatigue levels of the credit officers as neatly as some past studies have. For instance, in the Danziger et al. work, they were able to track fatigue levels throughout the day by noting when the decision makers (judges) took their breaks during the day. Specifically, they noted that after breaks (such as lunch), there was a temporary restoration of behavior corresponding presumably to reduced levels of fatigue, resulting in a scalloped shape function throughout the day. In the present research, without access to the specific times at which credit officers took their lunch (or other) breaks during the day, the authors had to simply extrapolate when most people took their lunch breaks to infer the presumed fatigue functions for these decision makers. These limitations are simply unavoidable, given the limitations imposed on those granting them access to the present data set, but they do limit the ability to draw clear conclusions about the role that fatigue presumably plays in the observed function of credit approvals. We can however safely assume that people are at their "best" (or at least are less fatigued) at the beginning of their work day and fatigue tends to increase as the day wears on.

Response: As we note in the response to Reviewer 2, this is indeed a major limitation of the data set. Again, we consider the applied relevance of our findings to be a major benefit: By being aware of the decision fatigue we documented, banks and other businesses can put in place measures to mitigate such effects, without necessarily having to specify whether each individual is affected equally. Thus, despite this limitation we believe that the results can be impactful.

Aside from these limitations in terms of the presumed fatigue levels of the decision makers, the ability to draw conclusions about the financial consequences of these decisions relies heavily on projections based on the banks' historic data of repayment rate of loans. This seems like a fair way to project the potential financial costs of decision fatigue here, but I will admit that I would have loved to have seen a more nuanced analysis of the applications that did or did not get approved as a function of time of day. I know the authors were not able to parse the data in this way, but it would be very informative to see what parameters of the loan applications get prioritized or ignored when decision makers are or are not fatigued. The authors have access to credit rating, loan amount, seniority level of the credit officer, but from a pragmatic standpoint, I was curious as to the potential for other factors (like SES or race) to infiltrate decisions at this point. That is, the authors conjecture that decision makers opt to follow their defaults more when fatigued, and that the default in this case is to be conservative and reject the application. But in other decision making contexts, we have seen that many other defaults can be operative when people are fatigued or stressed or distracted. People tend to rely on cultural stereotypes more, and thus may reject applications from minority group members (or companies or organizations) when fatigued than when at their full capacity. I know the authors are not able to explore these possibilities, but it is the case that there are other defaults that may be operative, if and when decision makers are feeling fatigued, that may govern their judgments and decisions under these conditions. It may not be that there is a drop in approval rate of any or all applications, but only of certain applications. And this point becomes particularly salient when the authors note that they could not

control whether the credit officers look at these applications in a random order or organize the applications in some way (such as doing the easier ones first and saving the more difficult ones for later, or vice versa).

Response: The Reviewer raises many fascinating questions that are very relevant to our work. In an ideal world we could test them all. Unfortunately, as the Reviewer also notes, this was simply not possible, not just because of the specifics of the data set that we had access to, but also because of privacy considerations typically present in the finance sector. For example, while SES and race are likely important factors, including this information in analyses could compromise the anonymity of the individuals who contributed the data, and could even lead to repercussions in their work place. So, even if such information was part of a given data set, we are unable to think of a way in which it could be analysed in an ethical manner.

These considerations, as well as many others raised by the reviewers, exemplify the limits of working with real-world data from the field. As noted above, we do not see any obvious solutions to this issue. We still believe that the findings are informative, especially in the context of the finance industry, which traditionally has been heavily guided by the “rational actor” model, with the current work suggesting that seemingly irrelevant factors can indeed have an important impact.

All this said, I think the present work is competently done and provides an intriguing new context in which to study the fundamental question of the (negative) consequences of decision fatigue. What seems to be an emerging pattern in studies like this is that decision fatigue (or depletion) effects appear to be more robust in real world settings, where you have real decision makers doing activities over time that cause fatigue. Thus, these studies have been helpful in reinforcing faith that depletion effects exist in the real world, despite the challenges (and failures to replicate) that researchers have observed in lab studies of this phenomenon. I really like the exploration of these questions in this particular financial decision making context, and sincerely appreciate the value of the present data set (with all its richness and external validity, despite the obvious and unavoidable limitations) to serve as the basis for this investigation. So from that perspective, I am very favorable disposed toward the publication of the present work.

Response: We appreciate the Reviewer’s positive evaluation of our work.

Still, though, the question to me is the level of contribution of the present work, over and above past real world demonstrations. The authors tout that this investigation affords the possibility of assessing the actual financial consequences of decision fatigue for the bank (in terms of profits or income). I appreciated their efforts in this regard, but I found that argument a bit hollow. The earlier studies which examined health care professionals’ compliance with recommended hand hygiene practices pretty directly illustrates the potential consequences of fatigue, in that risks of infection are greater when hands are not properly sanitized. Yes, they could not quantify the consequences in that study, but particularly in this day and age of COVID-19, anything that increases the threat of infection is something that we can all embrace as a clearly negative consequence of decision fatigue!

Response: Indeed, we agree that even without putting a price tag on decision fatigue, earlier work has illustrated the specific effects of decision fatigue. We now make this clearer in the paper.

Nonetheless, the present study tries to illustrate that point in a novel way, which I can admit is an advance over prior work. How much of an advance that constitutes is something that the AE will have to determine.

Response: We acknowledge that one could view the existing work as merely another demonstration of an effect that has already been observed on other contexts. However, as we note above, the finance section has been especially reliant on the “rational actor” model, and therefore the findings will likely come as a surprise to individuals in that industry, who are unlikely to be aware of other academic research.

Moreover, unlike other journals, Royal Society Open Science explicitly states in its scope that it “welcomes the submission of all high-quality science including articles which may usually be difficult to publish elsewhere, for example, replications or those that include negative findings.”

We greatly appreciate this publishing model, and believe it to be especially suitable to our study, given that (a) the data set has a number of limitations that are largely unavoidable given its real-world context, and (2) the work replicates earlier research on decision fatigue, and (3) indirectly also speaks to the controversy that has surrounded the phenomenon of ego-depletion. We therefore believe it will be a valuable addition to the literature, although, of course, it is ultimately for the editor to decide whether this is the case.

Overall, we thank the Reviewers for their valuable comments and believe the paper is much improved. If further changes would be considered useful, we would be glad to make them.

Sincerely,

Simone Schnall

Tobias Baer

Appendix C

Dear Professor Viding, thank you for your positive response and the additional feedback. Below we list the changes we made in response to the Reviewers' comments:

Reviewer: 1

The authors have addressed adequately all my comments. In the new version, the paper identifies better its contribution to the literature.

Response: We appreciate the Reviewer's positive evaluation of our work.

Only a minor comment. In my previous review I suggested to include the country of origin of the bank. I understand that it is not included by privacy reason. However, I also made the alternative suggestion of eliminating the wages from the paper, because they allow the identification of the country. The authors have clarified in the paper that original data are not in USD in order to show that the country is not US, but my point was different. The current footnote 3 still includes the monthly salaries identified through a Google search. I wanted to describe that that information allows to identify the country of origin, for instance for a person from that country who knows about the wages in the sector. If the authors want to keep the information confidential they could replace those numbers.

Response: This is an excellent point. We now amended Footnote 3 to protect the bank's identity.

The authors have included now the Gini coefficient of their model. Since the paper will be read by researchers from disciplines different from credit risk management, the authors should include a reference.

Response: We now added this reference.

Reviewer: 4

4) Regarding running a regression per credit officer. Part of my comment was to have some analysis to get a sense of the between credit officer variability, and to check that the effect is also valid when credit officers are considered as a random effect. The idea would be to run 30 regressions and to test the decision fatigue at the group level, that is to test whether the distribution of the 30 individual coefficients is different from zero. Instead of being a fishing expedition, this is a unique statistical test, which I still think would be useful. Moreover, it would be informative to plot the distribution of the coefficients for decision fatigue for the group. This would help the reader to know for example whether some credit officers are highly sensitive to decision fatigue and not others, or whether all of them are slightly sensitive to decision fatigue.

Response: While we appreciate the Reviewer's conceptual idea, we find it difficult to implement in our particular study context. First, "decision fatigue" is not a single variable in our model but we test for an effect of time of day - we take morning hours until 10:59am as the base line and then use the hourly intervals starting at 11, 12, 13, 14, 15, and 16 hours as well as the evening time from 17:00 onward as binary effects. We assume the Reviewer meant to analyse a particular hourly effect - such as the noon indicator, but we do not believe this would be appropriate. The reviewer suggests to run 30 separate regressions - one each for every credit officer - and to then test for the decision fatigue coefficients whether the distribution is different from 0. This approach collapses the 26,501 observations in our sample to just 30 coefficients, however. According to Cohen (Cohen, J. (1992). A Power Primer. Quantitative Methods in Psychology, 112(1), 155-159), for this type of analysis, one would need a sample size of at least 586 observations for a small effect, 95 observations for a medium effect, and 38 observations for a large effect (for a power of .80 and an alpha of .01). In other words, doing so would reduce the statistical power of a very large sample with 26,501 observations to the power of a very small sample with just 30 observations - which is insufficient for the analysis at hand. Furthermore, closer inspection of the data revealed that for some credit officers, at least one attribute of the loans had only one level, and this prevents the regression from being run. Overall, while we appreciate that the Reviewer made an intuitive suggestion, we do not find

it feasible to explore in our data set. We would, however, be happy to give this more thought if the Editor felt it was important to do so.

4) Regarding plotting the effect. I still believe that a plot with the whole data overlaying model predictions would be informative and unless I missed something such a figure has not been added to the revised version of the manuscript.

Response: In the last revision we had added Figure 1, which illustrates the effect of a typical loan (i.e., that takes into account various loan attributes). It would be misleading to instead plot only the means that do not reflect the various predictor variables included in the logistic regression. We therefore believe the current Figure 1 to be a more appropriate representation of the results.

7) Regarding goodness-of-fit metrics. The authors reported the Gini coefficient after having discarded the use of pseudo-R2. What method is used to generate a Gini coefficient (e.g., using the CAP curve, the AUC curve)? This metric also has its caveats. Could the authors also report the balanced accuracy? This is particularly suitable for imbalance data and easy to compute, and I do not see a good reason not to report it. This would help the reader to get a sense of the fit quality.

Response: Balanced accuracy is used in particular for highly imbalanced data which isn't the case here (about 40% of the sample are approved, about 60% are rejected) and it requires further assumptions (in particular, defining the cut-off value for labelling a prediction to be treated as a "yes" or "no"). The purpose of our research was to show whether the time-of-day effect is significant in order to test for decision fatigue. In the paper now cited in our article the balanced accuracy is deemed an inferior metric for evaluating a credit score, and we hence do not think it would be effective to side-track the reader on a methodology discussion that we consider largely tangential.

13) Regarding the day of the week effect, I noticed that another reviewer asked the same question. Reporting some statistics regarding the time of the week would be interesting while not being too costly. Without prior literature, a reasonable assumption is that decision fatigue would be stronger at the end of the week than at the beginning, in the same way that it is higher at the end of the day than at the beginning. Interestingly, weekend could be considered as long breaks and nights as shorter breaks and may even provide more evidence for a fatigue effect (rather than e.g., circadian rhythms).

Response: Although we did not have an a priori research hypothesis about the role of day of the week, the Reviewer's hypothesis is certainly plausible. One limitation of the data set, however, is that it included a public holiday (Friday, 25 March 2016, Good Friday) and the bank was closed. Presumably in anticipation of this holiday, we found that 424 cases had been actioned on the weekend prior to Good Friday, most of them on Saturday (only 12 on Sunday). In light of this, we added four day-of-the-week effects (coded as binary indicator variables) to our model: Tuesday, Wednesday, Thursday, and Friday/Saturday/Sunday. None of these were significant, however, as shown below. We therefore find no evidence for this intuitively plausible idea, but it certainly presents an intriguing avenue for future research.

	Estimate	Std. Error	z-value	p-value
Tue	0.014161	0.043096	0.329	0.742455
Wed	-0.063056	0.042849	-1.472	0.141132
Thu	-0.028298	0.043547	-0.650	0.515805
FrSaSu	-0.049583	0.047626	-1.041	0.297835

Reviewer: 3

I think that power calculations are important and should not be avoided. Thus, despite having large dataset I would encourage the authors to discuss the power of their statistical analysis. However, as I mentioned

earlier, one should be careful in applying formulas that assume independence between observations to longitudinal data.

Response: We appreciate the importance of sample size considerations, and now added the following section:

“Calculating the required sample size for logistic regression to achieve sufficient power is complex but a useful rule of thumb for the minimum sample size N is $N = 10 \times k / p$, with k being the number of model parameters and p being the smallest proportion of the two outcomes modelled (Peduzzi et al., 1996). With $N = 10 \times 51 / 39.95\% = 1,277$, our sample is roughly 20 times larger than what would be recommended using this rule of thumb.”

Furthermore, I think that the analysis of the relationship between seniority of the decision maker and loan restructuring decisions should be revised. Authors use the category “superintendent” as reference category, however, they only have one decision maker in that category. This can inflate the results. Thus, I would suggest to instead use a different reference category (for instance, junior analysts). Furthermore, it is good to note that there were only 30 unique decision makers and five seniority categories, and thus the conclusions regarding seniority should be made with big caution.

Response: Although we understand the desire to extract more information about specific work patterns or individuals, we hesitate to draw such inferences based on the relatively limited information in our study. We made this clear in Footnote 3.

Furthermore, it would also be valuable for readers to add a summary table with descriptive statistics of the dataset. In fact, authors attempted to describe the data by adding Figure 1. However, Figure 1 is not mentioned in the text at all. Besides, instead of simulated approval rates for an illustrative application, I think it is better to plot the actual (rather than simulated) approval rates for all applications considered within each time frame. In addition, it would be very informative to add the standard error bars as well as the number of applications considered within each time frame. It is currently unclear how many applications are considered within each time frame. This would give an idea of how the raw data looks like.

Response: As we note in the context of Reviewer 4’s comment, the reason for illustrating the effect on a typical loan (i.e., that takes into account various loan attributes) is that it would be misleading to instead plot only the means that do not reflect the various predictor variables included in the logistic regression. We therefore believe the current Figure 1 to be a more appropriate representation of the results, and we refer to it now on p. 5.

Overall, we thank the Reviewers for their valuable comments in the course of our revisions. If further changes would be considered useful, we would be glad to make them.

Sincerely,

Simone Schnall

Tobias Baer